# Associations Between Diabetes Mellitus and Neurodegenerative Diseases

**DOI:** 10.3390/ijms26020542

**Published:** 2025-01-10

**Authors:** Leszek Szablewski

**Affiliations:** Chair and Department of General Biology and Parasitology, Medical University of Warsaw, Chałubińskiego 5, 02-004 Warsaw, Poland; leszek.szablewski@wum.edu.pl

**Keywords:** type 1 diabetes mellitus, type 2 diabetes mellitus, neurodegenerative diseases (NDs), Alzheimer’s disease, Parkinson’s disease, Huntington’s disease, amyotrophic lateral sclerosis

## Abstract

Diabetes mellitus (DM) and neurodegenerative diseases/disturbances are worldwide health problems. The most common chronic conditions diagnosed in persons 60 years and older are type 2 diabetes mellitus (T2DM) and cognitive impairment. It was found that diabetes mellitus is a major risk for cognitive decline, dementia, Parkinson’s disease (PD), Alzheimer’s disease (AD), Huntington’s disease (HD), amyotrophic lateral sclerosis (ALS) and other neurodegenerative disorders. Different mechanisms of associations between these diseases and diabetes mellitus have been suggested. For example, it is postulated that an impaired intracellular insulin signaling pathway, together with hyperglycemia and hyperinsulinemia, may cause pathological changes, such as dysfunction of the mitochondria, oxidative stress inflammatory responses, etc. The association between diabetes mellitus and neurodegenerative diseases, as well as the mechanisms of these associations, needs further investigation. The aim of this review is to describe the associations between diabetes mellitus, especially type 1 (T1DM) and type 2 diabetes mellitus, and selected neurodegenerative diseases, i.e., Alzheimer’s disease, Parkinson’s disease, Huntington’s disease and amyotrophic lateral sclerosis. Suggested mechanisms of these associations are also described.

## 1. Introduction

Diabetes mellitus is a chronic disease, characterized by insufficient insulin production by the pancreatic β-cells as well as by insulin resistance. Insulin resistance is associated with the inability of skeletal muscle cells, adipose tissue and liver to respond adequately to insulin signaling. This disturbance of insulin signaling decreases uptake of glucose by cells due to DM [1], causing increased, prolonged blood glucose levels, called hyperglycemia. Impaired glucose metabolism, especially hyperglycemia, may stimulate serious damage to the heart, kidney, eyes and nerves. Increased levels of blood glucose may cause diabetic stroke, hypertension, cardiovascular disease, kidney failure and liver damage. Recently, neurodegeneration has also been added to this list [2]. There are two most common types of DM: type 1 diabetes mellitus and type 2 diabetes mellitus. T1DM is caused by destruction of insulin-producing pancreatic β-cells due to an autoimmune reaction. T1DM is commonly diagnosed in children and adolescents [3]. T2DM is characterized by insulin resistance and partial insulin deficiency. T2DM, accounting for more than 90% of patients, occurs mainly in adults [3]. It was observed that changes in human life styles and diet over the last decades cause increased prevalence of DM. In 2010, the world prevalence of diabetes mellitus among adults was 285 million, in 2019 the world prevalence of diseases was 463 million, and two years later, in 2021, the global estimate was over 536 million diabetic patients. It is suggested that 783.2 million people will be affected by 2045 [4,5].

The term “neurodegeneration” is used for the description of the “progressive, selective, neuronal structure loss, function and their population in the central nervous system (CNS)” [6]. Therefore, neurodegenerative diseases are associated with a progressive loss of structure and function of neurons. Progressive loss of specific neurons causes functional and cognitive deficits [7]. For example, in Alzheimer’s disease, the loss of pyramidal neurons in the Ammon’s horn is involved in the gradual deterioration of cognitive function of memory [8]. In patients with Huntington’s disease, neuronal loss of the dorsal striatum is observed. This neuronal loss results in involuntary movements (chorea), cognitive decline, as well as psychiatric symptoms [9]. Disturbances observed in patients with Parkinson’s disease are due to progressive loss of dopamine neurons in the substantia nigra and the degeneration of projecting nerve fibers to the striatum [10]. The pathophysiological mechanisms involved in the neuronal loss in the diseases mentioned are poorly understood. On the other hand, there are observed perturbations in glucose metabolism in all of these diseases [7], and there is a growing prevalence of dementia as the population ages. A characteristic feature of dementia is a diminished ability to lead a normal life on a daily basis, caused by cognitive impairment from poor memory, executive function and judgment, along with a decline in behavioral and social skills. Cognitive impairment starts with mild symptoms, but the level of disturbance progresses with time to frank dementia [11]. In 2019, dementia was diagnosed in 57.4 million people worldwide, and a rise to 152.8 million cases by 2050 has been suggested [12].

Several studies regarding the association between DM and dementia revealed an increased risk of dementia in diabetic patients. It was found that patients with T2DM treated with insulin have a relative risk as high as 4.3-fold for developing dementia [13]. Results obtained in another study revealed that patients with T2DM have a 65% increased risk for developing Alzheimer’s disease [14]. Investigations performed on Japanese-Americans in Hawaii revealed that diabetic patients had a 1.8-fold higher risk of vascular dementia. This pathology was also associated with low and high levels of fasting insulin, as levels of insulin is often, but not always, associated with diabetes mellitus [15,16]. A growing number of studies suggests the association between T2DM and neurodegenerative diseases, such as Alzheimer’s disease, Parkinson’s disease, Huntington’s disease, multiple sclerosis (MS), amyotrophic lateral sclerosis, etc. It is suggested that out of more than 100 neurodegenerative disorders, about 20% are associated with diabetes mellitus [17,18]. It is worth noting that elevated blood glucose levels also impair glial cells, such as astrocytes, which are involved in maintaining brain glucose levels. This role of astrocytes is due to neuron–astrocyte coupling. However, there are nine types of DM, but associations between DM and NDs were investigated and described mainly for T1DM and T2DM. Information on associations with the remaining types of DM is very poor and scarce.

As mentioned above, the prevalence of NDs will dramatically rise worldwide, due to increased life expectancy. These NDs mainly include Alzheimer’s, Parkinson’s and Huntington’s diseases and amyotrophic lateral sclerosis. The aim of this review is a description of the association between T1DM and T2DM and the above-mentioned neurodegenerative diseases, as growing numbers of studies revealed the involvement of DM in development of these diseases, and potential mechanisms of the correlation of DM and neurodegenerative disorders.

## 2. Types of Diabetes Mellitus

Diabetes mellitus is a metabolic disease, characterized by hyperglycemia and several other disturbances, such as insulin resistance, hyperinsulinemia, impaired insulin secretion, etc. There are nine types of DM, and there is also the diagnosed state of prediabetes. Prediabetes is a state when blood sugar is higher than it should be, but not high enough to diagnose it as diabetes. Blood glucose levels in prediabetes are between 110 mg/dL and 125 mg/dL (5.7–6.4 mmol/L). These patients are also defined by the presence of impaired fasting glucose (IFG) and/or impaired glucose tolerance (IGT). Patients with prediabetes exhibit obesity, especially abdominal or visceral, dyslipidemia with high triglycerides and/or low HDL cholesterol, and hypertension [19].

### 2.1. Type 1 Diabetes Mellitus

Type 1 diabetes mellitus is an autoimmune disease that is characterized by profound insulin deficiency due to destruction of pancreatic β-cells. T1DM requires the exogenous administration of insulin. Observed symptoms of this disease include the following: common ketoacidosis and hypoglycemia, severe hyperglycemia, normal or increased peripheral insulin sensitivity, low glucose levels, normal or high levels of glucagon and pancreatic peptide (PP), normal or low levels of glucose-dependent insulinotrophic polypeptide (GIP) and normal levels of glucagon-like peptide 1 (GLP1). The typical age of T1DM onset is childhood or adolescence [20]. Idiopathic type 1 diabetes mellitus is a form of T1DM that has no known etiology. Patients are diagnosed with insulinopenia and are prone to diabetic ketoacidosis (DKA), although without evidence of β-cell autoimmunity. This form of diabetes is mainly inherited, but without association with HLA [19].

### 2.2. Type 2 Diabetes Mellitus

Type 2 diabetes mellitus is associated with insulin resistance of peripheral tissues and cells and increased blood glucose levels (hyperglycemia). Chronic and prolonged hyperglycemia may cause damage to peripheral organs, such as nephropathy or retinopathy. Hyperglycemia also stimulates increased secretion of insulin by pancreatic β-cells, to obtain normoglycemia, resulting in hyperinsulinemia. Characteristic parameters of T2DM include the following: rare ketoacidosis and hypoglycemia, usually mild hyperglycemia, decreased peripheral insulin sensitivity, normal or decreased hepatic insulin sensitivity, high levels of insulin and PP, normal or high glucagon levels, normal or low GLP-1. Adulthood is the typical age of onset [20].

### 2.3. Gestational Diabetes Mellitus

Gestational diabetes mellitus (GDM) is diagnosed in women in the second or third trimester of pregnancy, without previous diabetes mellitus, and is associated with a degree of carbohydrate intolerance [21]. Obtained results revealed that women with a history of gestational diabetes mellitus have an approximately 7.4 times higher risk of developing postpartum T2DM, as compared to women who had normal glucose levels during pregnancy [22].

### 2.4. Type 3c Diabetes Mellitus

Type 3c diabetes mellitus (T3cDM) is also known as pancreoprivic or pancreatogenic diabetes. This DM is secondary to pancreatic disease, due to diseases of the exocrine pancreas [19]. T3cDM is associated with pancreatic diseases, such as pancreatic carcinoma, acute and chronic pancreatitis, cystic fibrosis, pancreatectomy and others [23]. For T3cDM, characteristic parameters include the following: rare ketoacidosis, mild hyperglycemia, common hypoglycemia, increased peripheral insulin sensitivity, decreased hepatic insulin sensitivity, insulin, glucagon, PP and GIP levels are low, and any typical age of onset [20]. Approximately 0.11% of patients with diseases of the exocrine pancreas have DM [24]. T3cDM affects approximately 9.2% of diabetic patients [25].

### 2.5. Maturity-Onset Diabetes of the Young

Maturity-Onset Diabetes of the Young (MODY) is characterized by the onset of hyperglycemia detected at an early age, before age 25 years, although diagnosis may happen at older ages. It affects 1–2% of diabetic patients. There are different subtypes of MODY (MODY 1 to MODY 14). The most commonly diagnosed types are MODY 1, MODY 2 and MODY 3 which account for 95% of cases [19,26].

MODY 1—Hepatic Nuclear Factor 4 α (HNF-4α) is similar to HNF-1 α, which is much less common [19,26,27].

MODY 2—Glucokinase. The glucokinase gene is involved in the regulation of the amount of insulin produced by the pancreatic β-cells in response to blood glucose levels. This type of MODY causes only minor changes in blood glucose levels, resulting in a rare development of diabetic complications [19,26,27].

MODY 3—Hepatic Nuclear Factor 1 α (HNF-1α). This subtype of MODY accounts for 70% of cases of MODY. It decreases the amount of insulin produced by the pancreas, causing diabetes. MODY 3 is often related to a very strong family history of diabetes. It may be misdiagnosed as T1DM, although patients are autoantibody negative [19,26,27].

MODY 4—PDX-MODY. PDX1 is a transcription factor that is involved in the exocrine and endocrine development of the pancreas and affects pancreatic development and expression of the insulin gene [19,26,27].

MODY 5—Hepatic Nuclear Factor 1 β (HNF-1β) is a very rare form of MODY. It is important to recognize, because MODY 5 may cause kidney problems in the form of renal cysts [26,27].

MODY 6—NEUROD1-MODY. NEUROD1 is a basic-loop-helix transcription factor associated with pancreatic and neuronal development. It is involved in maturation and maintenance of the pancreatic β-cell [26,27].

As a possible cause of MODY subtypes 7–14, eight genes have been suggested: Kruppel-like factor 11 (KLF11); carboxyl ester lipase; paired-box-containing gene 4 (PAX4); insulin (INS); B-lymphocyte kinase; adenosine triphosphate (ATP)-binding cassette, sub-family C (CFTR/MRP) number 8 (ABCC8); potassium channel, inwardly rectifying subfamily J, member 11 (KCNJ11); adaptor protein, phosphotyrosine interaction, PH domain, and leucine zipper containing (APPL1) [27].

MODY is often inherited as an autosomal dominant disease, in which at least 14 genes on different chromosomes are involved. Six genes encode proteins which correspond to MODY subtypes 1–6 [19,27].

### 2.6. Neonatal Diabetes

Neonatal diabetes is a diabetes that occurs before 6 months of age. Therefore, it is termed as “neonatal” or “congenital”. T1DM rarely occurs before 6 months of age. This type of DM may be either transient or permanent [19]. The first form, transient neonatal diabetes, is most often caused by overexpression of genes located on chromosome 6q24, and in about half of cases it is recurrent. The second form, permanent neonatal diabetes, is most commonly caused by autosomal dominant mutation in the genes which encode the Kir.2 subunit (KCNJ11) and SUR1 subunit (ABCC8) of the β-cell K_ATP_ channel and by mutation in gene *EIF2B1* affecting eIF2 signaling [28]. The second, most common cause of permanent neonatal diabetes is associated with mutations in the insulin gene, which are also predominantly inherited [19].

### 2.7. Latent Autoimmune Diabetes in Adults (LADA)

LADA, also termed 1.5 diabetes mellitus [29], accounts for 2–12% of all patients with adult-onset DM. Patients with LADA exhibit a slowly progressive form of autoimmune diabetes with immune markers of type 1 diabetes mellitus in serum. Diagnostic criteria for LADA are the following: adult-onset DM (>30 years at diagnosis), presence of autoantibodies associated with diabetes, and no requirement for insulin for least 6 months after diagnosis [29]. Several autoantibodies are detected in LADA patients, such as autoantibody against glutamic acid decarboxylase (GADA), which is the most sensitive marker used for screening. Other autoantibodies include those acting against islet cells (ICA), protein tyrosine phosphatase IA-2 (IA-2A), insulin (IAA), the islet-specific zinc transporter isoform 8 (ZnT8A), tetraspanin 7, and others [30,31,32]. There is also another, similar type of slowly progressive form of diabetes that is diagnosed in early-onset cases, called Latent Autoimmune Diabetes in the Young (LADY) [33].

### 2.8. Alström Syndrome

Alström syndrome (ALMS1) is a rare autosomal recessive genetic disorder. Its prevalence is from 1 in 500,000 to 1 in 1,000,000 [34]. It has been identified worldwide in approximately 1200 cases [35] and this syndrome affects both sexes equally [36]. It is due to mutation in the *ALMST1* gene, located on chromosome 2p13. Mutations in this gene are due to insertions, deletions and nonsense mutations, detected primarily in exons 8, 10 and 16. ALMS1 was first described in 1959 [37,38]. This gene encodes the ALMS1 protein that is found in centrosomes, basal bodies and cytosol [39]. This protein is involved in the function of cilia and its absence causes impairment of the formation of cilia [40]. The ALMS1 protein is also associated with homeostasis of energy metabolism, control of cell cycles, ciliary signaling pathways and intracellular trafficking [41]. Alström syndrome is also classified as a ciliopathy, because it causes abnormal cilia function and formation [42,43]. Diagnosis of Alström syndrome is difficult because some features begin at birth, whereas others are detected as the child develops [44]. Within and among families, symptoms differ between patients with this syndrome [44]. In children, the major phenotypes include cone-rod retinal dystrophy that begins in infancy and may cause juvenile blindness, sensorineural hearing, impairment obesity, insulin resistance, and congestive heart failure caused by dilated cardiomyopathy. In adolescence T2DM, hyperinsulinemia, hypertriglyceridemia, short stature, scoliosis, alopecia, and progressive renal, pulmonary and hepatic dysfunction are observed. Fibrosis of unknown etiology may develop in multiple organs. In adulthood, male patients exhibit hypogonadism and female patients, hyperandrogenism. Patients with Alström syndrome rarely exceed the age of 50 [36,39,44].

### 2.9. Wolfram Syndrome

Wolfram syndrome (WS) is a rare, progressive neurodegenerative disease. It is an autosomal recessive disorder [45,46]. This disease is also described by the acronym DIDMOAD (Diabetes insipidus, early-onset non-autoimmune insulin-dependent Diabetes Mellitus, Optic Atrophy and Deafness) [47]. According to the draft International Classification of Diseases (ICD-11), it is categorized as a rare specific diabetes mellitus, subcategory 5A16.1, Wolfram Syndrome [48]. Its global prevalence is one in 500,000 [49], or, based on other observations, it is estimated to afflict about 1 in 160,000–770,000 [50,51]. Patients with WS exhibit juvenile-onset diabetes mellitus, diabetes insipidus, optic nerve atrophy, sensorineural deafness, hearing loss, abnormal urinary tract function, neurodegeneration and neuropsychiatric disorders [43,52]. The first manifestation of WS is DM, usually diagnosed around age 6 [51]. The prognosis of WS is currently poor. Most patients die prematurely with severe neurological disabilities as a result of brain stem atrophy. The average life of patients with WS is approximately 30 years (full range 25–49 years) [51,53]. There are two phenocopies of WS: Wolfram Syndrome type 1 (WS1) and Wolfram Syndrome type 2 (WS2), which are due to homozygous mutations of *WFS1* and *CISD2* genes, respectively [52]. It is also the Autosomal Dominant Wolfram-like syndrome due to heterozygous mutations in the *WFS1* gene [52]. Wolfram Syndrome 1, commonly referred to as Wolfram syndrome, was first described in 1938 by Wolfram and Wagener in a family of eight siblings. Four had optic nerve atrophy and juvenile diabetes mellitus [54]. Its prevalence differs in dependence on the geographical region: from 1/54,478 in North-Eastern Sicily [55] to 1/700,000 in the general population and 1/500,000 in the pediatric population of the UK [48]. As mentioned above, WS1 is due to mutations in the *WFS1* gene, located on chromosome 4p16.1. It encodes the glycoprotein wolframin. To date, more than 200 mutations have been identified, mainly located in exon 8 [56]. It was found that nonsense and frameshift mutations had more severe phenotypic presentation in comparison with missense mutations [57]. Wolframin is a membrane glycoprotein located in the endoplasmic reticulum. It is involved in Ca^2+^ homeostasis and regulation of the endoplasmic reticulum stress response [58]. Wolframin is highly expressed in brain regions, such as the hippocampus, amygdala, allocortex, and olfactory bulb, as well as in pancreatic cells, heart, lung and placenta [47,59]. Mutations in the gene that encodes wolframin causes dysfunction of endoplasmic reticulum and mitochondria, resulting in apoptosis and cell death [60]. WS2 is a rarer form of WS due to mutation in the *CISD2* gene, located on chromosome 4p22–q24 [61]. It codes the zinc-finger protein called endoplasmic reticulum intermembrane small protein (ERIS). This protein is involved in the integrity of endoplasmic reticulum and mitochondrial membrane, as well as in the functional interaction between these two cellular compartments [62]. ERIS is expressed in several tissues, such as brain and pancreas [59,63].

## 3. Disturbances Associated with Diabetes Mellitus

As mentioned above, DM is a metabolic disease in which metabolism of carbohydrate, protein and fat is altered. There are several disturbances associated with particular types of DM. Sometimes it may be problematic, whether the impairment or pathology in question is a cause or an effect of diabetes mellitus. In this chapter, mainly disturbances associated with T1DM and T2DM will be described.

### 3.1. Insulin Resistance and Hyperinsulinemia

In physiological concentrations, insulin plays an important and positive role in the whole body. It is also involved in several processes taking place in the nervous system. The hormone controls cell growth, regeneration, and the development of the brain and its functions. The intracellular insulin signaling pathway regulates processes associated with mitogenesis, autophagy, apoptosis and others (Figure 1).

There are several descriptions of insulin resistance: “a state of reduced responsiveness in insulin-targeting tissues to physiological levels of insulin” [65] or “a state of a cell, tissue or organism in which a greater than normal amount of insulin is required to elicit a quantitatively normal response” [66], or “an inability of some types of tissues to respond to normal insulin levels, and thus, higher than normal levels of insulin are required to maintain the normal function of insulin” [65]. There are a few types of insulin resistance, such as Danohue syndrome, Rabson-Mendenhall syndrome (RMS), type A insulin resistance syndrome (TAIRS), type B insulin resistance syndrome (TBIRS), type C insulin resistance syndrome (HAIR-AN), lipodystrophies and other severe insulin resistance syndromes [67]. In patients with T2DM, insulin resistance and hyperinsulinemia are commonly diagnosed. In patients with T1DM, hyperinsulinemia may be misdiagnosed due to administration of exogenous insulin. Clinical observations revealed insulin resistance in patients with T1DM [68], a prominent feature in adolescents and adults [69,70]. Insulin resistance in these patients is detected mainly in liver, peripheral and adipose tissue [71]. Prolonged elevated blood glucose levels in patients with T2DM may be due to decreased efficiency of insulin in cells and reduced utilization of glucose by cells. Therefore, to obtain normoglycemia, pancreatic β-cells secrete higher amounts of insulin, resulting in hyperinsulinemia. The compensatory mechanism is associated with hyperplasia of pancreatic β-cells and overproduction and hypersecretion of insulin [72]. Constant overproduction and hypersecretion of insulin causes dysfunction of β-cells, resulting in the appearance of T2DM [73]. Increased blood insulin levels impair cellular function. Therefore, the use of the term “insulin toxicity” is suggested [74]. For example, high fasting insulin levels in children and adolescents cause higher weight gain in later years [75]. The mechanisms of insulin resistance are multifaceted and are associated with low-grade inflammation [76], glucose metabolism alteration, causing hyperglycemia and advanced end products formation [77], increased levels of free fatty acids and triglycerides [78,79], oxidative stress [80], mitochondrial dysfunction [81], endoplasmic reticulum stress [82] and reactive nitrogen species (RNS) [83,84]. Cellular oxidative stress generates free radicals which are highly reactive molecules and their increased levels may damage protein, lipids and nucleic acids [85]. Hyperinsulinemia due to insulin resistance is also associated with diseases such as non-alcoholic fatty liver disease (NAFLD) [86], polycystic ovary syndrome [87,88], Alzheimer’s disease [89,90] and cardiovascular and cerebrovascular diseases [91,92,93]. Insulin is known as a factor that may stimulate development and progression of cancers. Obtained results revealed an association between insulin resistance and hyperinsulinemia with certain cancers, such as breast, colorectal, prostate, pancreatic, adrenocortical, endometrial and other cancers [67].

### 3.2. Hyperglycemia

Hyperglycemia is one of the main pathologies observed in patients with T2DM. It occurs when the fasting blood glucose level is higher than 125 mg/dL or 180 mg/dL postprandial [94,95]. Decreased physical activities and increased obesity may stimulate development of hyperglycemia. Factors which are involved in the onset and progression of hyperglycemia are, for example, dysfunction of the islets of Langerhans, decreased secretion of insulin, impaired glucose utilization, and insulin resistance [96,97,98]. Basal hyperglycemia is caused by a lower insulin-to-glucagon ratio resulting in increased production of glucose by the liver. Postprandial hyperglycemia is due to decreased levels of blood insulin or action caused by reduced utilization of glucose in peripheral tissues [99]. Postprandial hyperglycemia depends on several factors, such as quantity and composition of food, content of carbohydrate as well as production of insulin and inhibition of glucagon secretion [96,97,98]. A positive correlation between glucose levels and risk of cardiovascular disease has been found [100]. Prolonged hyperglycemia contributes to development of the onset of complications, such as diabetic ketoacidosis and hyperglycemia hyperosmolar syndrome [101]. A phenomenon known as the Warburg Effect, characteristic for cancer cells, causes cancer cells to need an increased uptake of glucose to obtain sufficient energy. Therefore, hyperglycemia may provide more glucose for survival and proliferation of cancer cells. The synthesis of protein and DNA in cancer cells depends on glucose metabolism. High blood glucose levels have positive effects on cancer growth and metastasis [102]. Hyperglycemia, associated with high-energy utilization and toxicity of glucose, influences the synthesis of reactive oxygen species (ROS) and damages DNA, causing oxidative stress and the inflammatory response. A high blood glucose level is also involved in the production of advanced glycation end products (AGEs), generates ROS and increasing oxidative stress, increased pro-inflammatory signalling [103,104].

### 3.3. Oxidative Stress

Oxygen is an element associated with the aerobic cellular process. It may generate highly reactive and potentially toxic free radicals. In the outer orbit, oxygen contains two unpaired electrons. It can be changed into a highly reactive oxygen singlet with one unpaired electron. These changed chemical species are referred to as reactive oxygen species. The generation of ROS involves enzymes, such as nicotinamide adenine dinucleotide phosphate (NADPH), oxidase (NOX) and xanthine oxidase (XO) [105]. There are also several nitric oxide synthases (NOS)-derived metabolites which are referred to as reactive nitrogen species (RNS) [106]. Endogenous ROS/RNS chemicals may be predominantly generated by mitochondrial and some non-mitochondrial enzymes. Free radicals generated by mitochondria are an effect of the oxidative phosphorylation at the respiratory complexes I and II; however, ROS may also be generated in mitochondrial matrix by flavin-dependent enzymes. RNS is an effect of the nitric oxide synthase (NOS)-derived nitric oxide (NO) reaction with ROS [105]. At physiological concentrations, ROS/RNS play an important role in several cellular pathways and processes, such as stressor responses, cell survival and inflammation [107]. Under physiological conditions, ROS/RNS are scavenged by enzymes, such as superoxide dismutase (SOD) and catalases, which are involved in enzymatic antioxidant defense mechanisms. This role may also be filled by natural antioxidants, such as glutathione (GSH) and vitamin C. Overproduction of these reactive molecules may cause damage of the cellular macromolecules, such as DNA, proteins and lipids [105]. Presence of ROS in high levels may cause oxidative stress (OS) resulting in cell death. These reactive molecules are also involved in the development and progression of many diseases, such as cardiovascular disorders, allergy, and cancer, as well as neurodegenerative diseases, such as Parkinson’s disease and Alzheimer’s disease [89,104,107,108,109]. The association between the reactive oxygen species and neurodegenerative disorders is well established in diseases such as Alzheimer’s, Parkinson’s and Huntington’s diseases, amyotrophic multiple sclerosis and Friedrich’s ataxia [109,110]. Chronic hyperglycemia observed in diabetes mellitus is associated with generation of OS, resulting in neuronal damage [111]. Also, mitochondrial dysfunctions are associated with the development of neurodegenerative diseases and diabetes [112].

## 4. Diabetes Mellitus and Alzheimer’s Disease

Type 2 diabetes mellitus and cognitive impairment are common conditions, diagnosed in persons aged ≥60 years. Obtained results revealed that approximately 18–20% of individuals after that age have DM; in approximately 19% of cases, mild cognitive impairment in multiple domains is observed, and approximately 6% of patients have some dementia [113]. The number of persons with mild cognitive impairment and dementia in addition to DM increases with age. Several studies have revealed an association between diabetes mellitus, cognitive impairment, dementia and neurodegenerative diseases, such as Alzheimer’s, Parkinson’s and Huntington’s diseases, etc. For example, Parkinson’s affects 1–2% of the population aged ≥65 years, while Alzheimer’s is the sixth most-common cause of death in the United States [114,115].

### 4.1. Associations of Type 2 Diabetes Mellitus with Alzheimer’s Disease

The most common cause of dementia, which accounts for 60–70% of all cases, is Alzheimer’s disease. It affects approximately 15% of persons of age ≥65 years, and amongst those aged ≥85 years—approximately 45% [116]. There are over 100 types of dementia. But the most well-known form of dementia is Alzheimer’s disease (AD), which accounts for 50–75% of all cases of dementia [117]. Research suggests that diabetes mellitus has the most influence on the development of AD [118,119]. It was found that AD may cause disturbances at the biochemical, histopathological and molecular levels, and can be considered one of the complications of type 2 DM [120]. The risk of developing Alzheimer’s disease is increased by 50–60% in patients with T2DM [121]. Other epidemiological studies revealed that the risk of Alzheimer’s disease is increased by at least 2-fold in patients with T2DM [122]. Therefore, it is suggested that a correlation exists between the pathology of Alzheimer’s disease and type 2 diabetes mellitus. Based on meta-analysis of diabetes mellitus cases and the risk of dementia, including AD and vascular dementia, an increase of 73% was found in the risk of dementia and 56% risk for developing AD in patients with diabetes [123]. These results suggest that diabetes mellitus increases susceptibility of AD, therefore the use of a new term for AD is suggested: type 3 diabetes mellitus (T3DM) [124,125,126,127].

Alzheimer’s disease is a progressive neurodegenerative disorder [128]. Patients with AD exhibit severe and progressive impairment of cognitive function [129], associated with memory loss and language problems [130]. There is also observed non-cognitive dysfunction (executive) that may be followed by behavioral disorders, such as agitation, aggressiveness and depression [129,130,131]. The brain areas most affected by the disease are the neocortex and hippocampus [129]. In these areas, loss of neurons is observed, resulting in their atrophy [132]. There are clinically two subtypes of Alzheimer’s disease [133]. About 95% of patients with AD are aged ≥65 years. These patients are diagnosed with “late onset” or “sporadic AD” (sAD). In 5% of patients with AD, the disease is due to a rare genetic mutation. These patients are diagnosed with “early onset” or “familial AD” (fAD). Symptoms of disease in patients with fAD are observed in younger persons—thirties, forties and fifties [134]. Symptoms observed in fAD are due to mutations in three genes: amyloid precursor protein (APP), presenilin-1 (PS-1) and presenilin-2 (PS-2). It is suggested that there are also other mutations associated with fAD, but these are yet unknown [133]. The genetics of sAD is more complex [135]. It was found that in the case of sAD, the epsilon 4 allele of the apolipoprotein E (ApoE4) gene is a significant risk factor for the development of sAD. Research has shown that possession of two copies of the ApoE4 gene increases the risk of Alzheimer’s disease by 12-fold. The risk of AD is increased 4-fold by the presence of one copy [136]. The carriers of this gene are 50–60% of all individuals. Therefore, it is postulated that other factors influence the risk of developing AD [137]. Results obtained in a cohort study revealed that T2DM is associated strongly with sAD after adjustment for sex and age. The authors also suggest that the link between type 2 diabetes mellitus and late-onset AD is partly associated with cerebrovascular pathology [138]. Observations of an elderly Chinese population with mild cognitive impairment showed that T2DM stimulates the progression of Alzheimer’s disease, while in age-matched controls, no association was observed [139]. Similar results were also obtained in earlier studies [140,141]. The link between the ApoE4 genotype and T2DM has also been investigated. Unfortunately, the results obtained are controversial. For example, in one study, results revealed an increased risk of sAD due to ApoE4 in persons with T2DM [15], while in another study, an association between T2DM and risk of dementia was confirmed only in ApoE4 non-carriers [142].

### 4.2. The Pathogenesis of Alzheimer’s Disease

Researchers have suggested the theory that cause and effect of AD is associated with abnormal protein aggregation [143,144]. The proteinopathy symptoms observed in AD consist of two types of misfolded protein in the brain. One is neurofibrillary tangles (NFTs) containing hyperphosphorylated tau, and the second is cellular plaques with aggregated amyloid β (Aβ) peptides [145,146].

#### 4.2.1. Accumulation of Amyloid β Peptides

The pathogenesis of Alzheimer’s disease begins when synaptic function is impaired. This pathology is due to the accumulation of Aβ that is produced from the amyloid precursor protein (APP). APP is a transmembrane protein that is expressed in many tissues, including in the central nervous system. The main sites of its concentration are the synapses of neurons. It is a cell surface receptor that plays a role as a regulator of synapse formation, neural plasticity, antimicrobial activity and export of iron [89]. There are two different pathways in which APP is processed. The first pathway, in which 90% of APP is processed, is the non-amyloidogenic (nonplaque-forming or secretory) pathway. The remaining 10% of APP is processed by the amyloidogenic processing of APP due to regulation of its phosphorylation. Based on these observations, it is clear that the accumulation of pathologic Aβ may be due to defects in insulin signaling [147]. In the non-amyloidogenic pathway, α-secretase cleaves APP, resulting in a C-terminal fragment α (CTFα, C83) and N-terminal fragment α (sAPPα). Then, γ-secretase cleaves CTFα, obtaining a smaller C-terminal fragment (C3) [148,149]. In the amyloidogenic pathway, a process that takes place within the endosome, β-secretase cleaves APP into a smaller N-terminal (sAPPβ) fragment and a longer C-terminal fragment. The C-terminal fragment contains the full amino acid sequence (CTFβ, C99). Cleavage of CTFβ due to γ-secretase causes the formation of an APP intracellular cytoplasmic domain (AICD) and fragment of Aβ. The resulting AICD and Aβ are released into the extracellular environment, where the Aβ fragments form oligomers, prefibrillar aggregates and fibrils, resulting in the formation of Aβ plaques [148,149]. Chains of Aβ peptide may contain different numbers of amino acids: 38 (Aβ_38_), 40 (Aβ_40_) and 42 (Aβ_42_). Increased expression of APP, together with altered proteolysis, causes the accumulation of 40- and 42-amino acid Aβ peptides. In fAD, the role of γ-secretase is changed. This change is due to mutations in the APP, PS1 and PS2 genes, as well as the inheritance of apolipoprotein Eε4 (ApoE-ε4), resulting in increased production of Aβ_42_ [150]. In healthy subjects, Aβ is released out of the neurons and its levels are controlled by local proteases [151]. Inappropriate siting of all the cleavage may increase levels of Aβ_42_, which is a neurotoxic isoform. Aβ peptides aggregate into oligomers and are involved in the creation of fibrils and amyloid plaques. Plaques block the signaling pathway and cell connections, resulting in cell death. Insulin plays an important role in the metabolism of APP which controls and regulates the balance between anabolism and catabolism of Aβ. Decreased levels of insulin or the lack of insulin’s action increases formation of neurofibrillary tangles, resulting in oxidative damage of neurons. Decreased levels of insulin also increase levels of Aβ and formation of amyloid plaques in the brain [149,152]. Insulin and Aβ are substrates for insulin-degrading enzyme (IDE). Insulin increases the levels of IDE, therefore, impairment of the insulin signaling pathway reduces degradation of Aβ [153].

#### 4.2.2. Tau Hyperphosphorylation

The tau protein is a neuronal microtubule-associated protein, the presence of which is detected in axons [154]. Six isoforms of this protein have been described in the human brain, due to alternative splicing [155]. It is classified as a microtubule-associated protein (MAP) [156]. The main function of tau is its involvement in the assembly and stability of microtubules. Microtubules are involved in several processes, such as morphogenesis, cell division and intracellular trafficking. In neurons, it is associated with synapses and nuclei [157], from where tau is also released into the extracellular space [158]. The function of released tau is unknown [159]. Regulation of the tau protein activity is due to its phosphorylation. Tau contains more than 85 phosphorylated or phosphorylable sites; about 80 Ser/Thr and 5 Tyr phosphorylation sites [160,161,162]. To maintain neuronal homeostasis, kinases and phosphatases are involved in the regulation of balance between tau phosphorylation and dephosphorylation. Imbalance between these phosphatases and kinases may cause hyperphosphorylation. Phosphorylation of tau is due to kinases, such as GSK-3β, CDK5, MARK, PKA, PKB/AKT and MAPK, including ERK ½, c-Jun N-terminal kinase (JNK) and p38 [163,164]. The expression of the *Tau* gene is regulated by insulin [165]. Insulin regulates phosphorylation of tau due to inhibition of GSK-3β, which is a key kinase associated with phosphorylation of tau. It was found that it phosphorylates tau at more than 30 sites, so it may play a key role in the development of Alzheimer’s disease and NFT [154]. Regulation of GSK-3β by insulin is due to the PI3-K/AKT/GSK-3β intracellular insulin signaling pathway [149,166]. Impairment of the insulin signaling pathway, caused by inhibition of the PI3-K/AKT pathway, increases activation of GGSK-3β [165]. Protein phosphatases 1 (PP1) and PP2 are also involved in the insulin signaling pathway [149,166]. It has been found that in diabetes mellitus, PP2 is downregulated [167,168]. Caspases (proteolytic enzymes) cleave tau protein into tau fragments. These fragments stimulate the formation of tau fragments, resulting in initiation of the process of tau aggregation [169,170]. Diabetes and hyperglycemic conditions activate caspases [171,172]. An impaired brain insulin signaling pathway reduces the activity of AKT, causing increased activity of GSK-3β. This causes hyperphosphorylation of tau, resulting in the formation of tau fibrils [173]. In brains of patients with AD, tau protein is three times more hyperphosphorylated compared to tau phosphorylation in brains of healthy subjects [148]. Insulin also influences expression of tau, therefore, a decreased insulin signaling pathway impairs expression of tau [174]. Investigations of brains in patients with AD revealed that activated caspases cleave tau protein at several sites. Caspase-3 cleaves the C-terminal of tau, giving rise to Asp421 residue. It has a high prosperity of aggregation. Therefore, it is involved in the development of neurofibrillary pathology in brains of patients with AD. These observations were also confirmed in animal studies, [164,175,176].

### 4.3. The Role of Type 2 Diabetes Mellitus in the Development of Alzheimer’s Disease

Several pathologies are observed in type 2 diabetes mellitus, such as insulin resistance, hyperinsulinemia, hyperglycemia, etc. These disturbances may be associated with the development of Alzheimer’s disease (Figure 2).

#### 4.3.1. Insulin Resistance and Alzheimer’s Disease

Insulin resistance is defined as “reduced sensitivity in body tissues to the action of insulin” [178]. It may be also treated as disturbances in neuroplasticity or release of neurotransmitters in neurons, resulting in decreased ability to regulate the metabolism or impaired cognition and mood [179]. T2DM causes metabolic disorders in the peripheral tissues, which are similar to the disorders observed in brains of patients with AD. Based on this observation, researchers postulate that AD is a form of T2DM associated with the brain [180], therefore AD may be named T3DM [120].

Insulin resistance is associated with an impaired insulin signaling pathway, caused by defects in function of the insulin receptor. The PI3-K/AKT pathway, in which GSK-3β is a component, inhibits GSK-3β in normal conditions. Impairing this pathway due to insulin resistance causes activation of GSK-3β, resulting in enhanced phosphorylation of tau [89,165,181,182]. Protein phosphatase PP2A, which plays an important role in homeostasis of tau phosphorylation, is downregulated in T2DM. Insulin resistance may increase phosphorylation of tau due to downregulating PP2A [167]. Impairment of the insulin PI3-K/AKT signaling pathway due to insulin resistance may decrease regional cerebral blood flow, resulting in decreased brain supply of oxygen, glucose and nutrients [183,184]. Neurons in the brain need high energy, therefore they consume up to 80% of oxygen in the brain [185]. This observation suggests that neurons are highly sensitive to disturbances of glucose metabolism and dysfunction of mitochondria. Research has shown a common association of neurodegenerative disorders with an impaired uptake of glucose and perturbations in glucose-related pathways, such as glycolysis, pentose phosphate pathway (PPP), tricarboxylic acid (TCA) cycle and oxidative phosphorylation (OXPHOS) [117,185,186]. Activation of the PI3-K/AKT pathway by insulin regulates vasodilation and vasoconstriction [187]. Stimulating endothelial nitric oxide synthase (eNOS) causes the production of nitric oxide (NO), resulting in vascular relaxation [187]. Insulin resistance, causing impairment of the vasodilatory PI3-K pathway and decreased production of NO, results in vasoconstriction. Impairment of this signaling pathway results in decreased availability of nutrients to the brain, causing an increase of oxidative stress and production of reactive oxygen species, leading to increased inflammatory response [183]. Insulin resistance also impairs the ability to inhibit glucose output, causing glucose toxicity. Also, glucose uptake and glucose metabolism are affected in the insulin-resistant state, resulting in impacted neuron function [188]. Insulin resistance may also cause other pathologies associated with T2DM, such as hyperglycemia, hyperinsulinemia, oxidative stress, etc.

#### 4.3.2. Hyperglycemia and Alzheimer’s Disease

Elevated glucose levels—hyperglycemia—is a common pathophysiological feature of T2DM that affects diabetic patients. It may be an effect of insulin resistance. Hyperglycemia may cause neurotoxicity, a process associated with increased glucose levels in neurons, causing neuronal damage and neuropathy [189]. The brain is a strictly glucose-dependent organ that needs a constant supply of glucose for maintaining its physiological functions [190,191]. This energy obtained from glucose is necessary for several processes, such as maintaining the resting membrane potential, action potential, postsynaptic potential in neuronal cells, as well as the synthesis and release of neurotransmitters [192,193]. Research has revealed that constantly elevated blood glucose levels, as is observed in T2DM, increase the risk of developing dementia [194] and, as observed in animal studies, cognitive dysfunction and memory deficits. These studies showed AD-like biochemistry changes in animals. Increased levels of hyperphosphorylated tau protein and decreased levels of synapse-associated protein were observed in animal models [195]. Other animal studies revealed that a high-sugar or high-fructose diet causes cognitive impairment, loss of memory and deposits of Aβ protein [196]. Other animal studies, performed on a rat model of diabetes, revealed that hyperglycemia causes cognitive impairment. It was also found that microglia were activated and continuously released pro-inflammatory cytokines [197]. Activated microglia were also detected in the brains of diabetic patients [90,198]. High glucose in the endothelial cells of the cerebral vascular system causes damage to neurons. Hyperglycemia increases advanced end products, causes an imbalance in the formation and destruction of reactive oxygen species, and impairs the intracellular signaling pathway [199]. High glucose levels may induce tissue damage, such as to the polyol pathway, amplifies glucose influx, activates protein kinase C and its isoforms, damages the hexoamine pathway [200], and converts glucose and fructose to sorbitol using nicotinamide-adenine dinucleotide phosphate (NADPH) that plays an essential role in the regeneration of endogenous antioxidant glutathione [201]. Hyperglycemia depletes antioxidants and causes the accumulation of free radicals which may activate redox-sensitive genes, thus changing the redox potential of cells as well as stimulating tissue damage [202]. The advanced end products are formed from glucose. In patients with T2DM, increased formation of ROS stimulates the production of AGEs [203]. Postmortem studies on brain samples showed that patients with T2DM and AD have higher levels of AGEs and activated microglia in comparison to patients with Alzheimer’s disease but without T2DM [204]. Production of ROS due to AGEs stimulates APP-related signaling pathways, resulting in formation and accumulation of Aβ [199,205]. Results obtained by other research also suggest that transient hyperglycemia, observed in T2DM, may be involved in the formation of Aβ plaques [133]. An increased accumulation of AGEs in the brain of diabetic model animals showed that products of AGEs disturb the removal of Aβ_42_, inducing aggregation of Aβ in the brain [206]. For more details, see [207].

#### 4.3.3. Hyperinsulinemia and Alzheimer’s Disease

Research has shown that hyperinsulinemia may influence the risk of AD. Impairment of the intracellular insulin signaling pathway plays a critical role in the pathogenesis of Alzheimer’s disease, as insulin may indirectly enhance the cleavage of APP due to controlling the γ-secretase complex, increasing the level of Aβ [208]. Hyperinsulinemia is an effect of insulin resistance. In an insulin-resistant state, the sensitivity of the body cells to insulin is decreased. Regulation of blood glucose levels is decreased, causing hyperglycemia. As a compensatory action to maintain normal blood glucose levels, pancreatic β-cells secrete more insulin [209]. Association of hyperinsulinemia with an increased risk of AD is caused by modulation of the toxicity of Aβ. As mentioned earlier, Aβ protein is degraded by an insulin-degrading enzyme that also degrades insulin [210]. In T2DM, peripheral hyperinsulinemia causes increased levels of insulin. Insulin acts as a substrate for IDE, and therefore inhibits the degradation of Aβ, resulting in accumulation of insoluble plaques. Animal studies have shown that deletion of the IDE gene (IDE -/-) in mice decreases by 50% Aβ clearance in brain homogenates [211]. Loss of IDE in mice causes increased cerebral accumulation of endogenous Aβ, enhanced by hyperinsulinemia and glucose intolerance, pathologies observed in T2DM [133]. Elevated levels of insulin also impair transport of amyloid precursor protein. Studies in vitro revealed that in cases of overexpression of βAPP in N2a cells and primary neuronal cultures, insulin reduces the intracellular accumulation of Aβ due to stimulation of the transport of βAPP/Aβ from the trans-Golgi network to the plasma membrane [212]. Obtained results also showed that insulin increases the concentration of extracellular Aβ, thereby modulating the trafficking of APP. The influence of the receptor tyrosine kinase/mitogen-activated protein (MAP) kinase pathway on regulation of intracellular transport of Aβ is also suggested [213].

#### 4.3.4. Oxidative Stress and Alzheimer’s Disease

Reactive oxygen species are groups of atoms which have odd, unpaired electrons. ROS causes oxidative stress, influencing the pathophysiology of neurodegenerative disorders. This may be enhanced by dysfunction of mitochondria and reduced expression of antioxidant genes [214]. Examples of free radicals are superoxide, oxygen, hydroxyl, alkoxy, peroxyl radicals, nitric oxide and nitrogen dioxide [215]. Non-radical ROS include hydrogen peroxide, hypochlorous acid and nitrogen compounds [109]. Oxidative stress may be induced by several pathologies, for example, by insulin resistance. It may be an effect of dysregulation of carbohydrate and lipid metabolism, increased activation of GSK-3β, disturbance in cell survival and anti-apoptotic signaling, or impaired energy balance and mitochondrial dysfunction. Hypoxia and ischemia also promote oxidative stress [216]. The brain is a highly metabolically active organ. Therefore, it produces large amounts of ROS. The brain is protected against oxidative stress by the action of antioxidants [217]. Generation of ROS and RNS may also be increased by oxidative stress. These reactive oxygen and nitrogen species react with macromolecules, such as lipids, proteins and nucleic acids, causing neurotoxic effects [218]. They are also involved in mitochondrial dysfunction in AD [219]. Oxidative stress occurs when cellular metabolism is greater than the capacity of antioxidants. It may occur also when excessive amounts of ROS and RNS are produced and these amounts are too large to be removed [117]. The capacity of cerebral antioxidant defenses, described as overproduction of oxidative stress, was detected in patients with type 2 diabetes mellitus [220,221]. In diabetic patients, hyperglycemia stimulates peroxidation of low-density lipoprotein in a superoxide-dependent manner, influencing the generation of free radicals [221]. As mentioned above, oxidative imbalance causes several pathologies, such as activations of microglia and astrocytes, deformations of protein and damage to DNA [222]. There are many studies which describe the associations of oxidative stress in pathogenesis of DM, with changes in enzymatic systems, peroxidation of lipids, impaired metabolism of glutathione, decreased levels of vitamin C, and so on. For more details, see [6].

Oxidative stress is also associated with mitochondrial dysfunction. As mentioned earlier, the brain has a high energy demand. Therefore, mitochondria, which play a role as a bioenergetic source, are important in the maintenance of neuronal functions [223]. Mitochondrial energy generation is due to processes such as glycolysis, the electron transport chain and oxidative phosphorylation. Mitochondria are the main generators of ROS, which are generated as a side product of the metabolic processes mentioned. Generated ROS are scavenged by cellular antioxidant defense mechanisms. Hyperglycemia occurring in diabetic patients causes overload of the mitochondrial metabolic processes, generating excess ROS [111,224]. Mitochondrial dysfunctions due to hyperglycemia may cause neuronal death via apoptosis and/or necrosis [225,226]. Dysfunctional mitochondria produce less ATP, but generate more ROS [227]. Dysregulated mitochondria may be involved in disturbances in calcium homeostasis [228], resulting in apoptosis and memory impairment [229,230,231]. Excessive ROS contributes to the generation of RNS. Reactive nitrogen species bind with molecular oxygen to the cytochrome *c* oxidase, causing impairment of mitochondrial function and inhibition of ATP synthesis [232,233]. Oxidative stress also stimulates mutations in mitochondrial DNA (mtDNA). The accumulation of these mutations impairs the electron transport chain, which may affect production of ATP, destroy brain function and increase the risk of Alzheimer’s disease occurrence [229,230,231]. It is noteworthy that the mutations mentioned are also detected in patients with T2DM. There are also proteins, such as peroxisome proliferator-activated receptor gamma coactivator 1α (PGC-1α), nuclear respiratory factor 1 (NRF1), nuclear respiratory factor 2 (NRF2) and mitochondrial transcription factor A (TFAM), which are associated with processes of mitochondrial biosynthesis. These proteins are also involved in neurodegenerative disorders [234]. In patients with initiated AD, the expression of these genes is decreased [234]. Mitochondrial dysfunction is involved in neuronal death, an important cause of neurodegeneration. Therefore, it is suggested that impaired mitochondrial function is a connecting link between Alzheimer’s disease and diabetes mellitus [235,236]. However, several investigations revealed mitochondrial dysfunction in AD, but the specific role of this organelle in Alzheimer’s disease is still controversial. There are two main hypotheses [112]. The first may be named the “mitochondrial cascade hypothesis”. According to this hypothesis, mitochondrial dysfunction is involved in AD [237]. The second may be named the “MAM hypothesis”, which suggests that impaired mitochondrial functions are the downstream consequence of mitochondria-associated endoplasmic reticulum membranes (MAMs) dysfunctions, which is the primary disturbance in AD [238]. A very important problem under debate is this: are mitochondrial dysfunctions the cause of Alzheimer’s disease? However, a widely-accepted suggestion is that mitochondrial dysfunction is involved in the development of AD. On the other hand, there are results from other studies which indicate that dysfunctional mitochondria in diabetic patients may also interact with impaired insulin signaling in these patients, stimulating the development of Alzheimer’s disease [239,240].

#### 4.3.5. Inflammation and Alzheimer’s Disease

Another pathogenic mechanism which may link T2DM and AD is inflammation. Its role in the development of Alzheimer’s disease is well-documented [183]. Research has shown that hyperinsulinemia, often diagnosed in diabetic patients, stimulates inflammatory responses in the central nervous system [241]. Diabetic patients have elevated levels of circulating cytokines and chemokines [242], associated with hyperinsulinemia [6], in which increased cerebral levels are detected [243]. Dementia and its mechanisms are associated with peripheral and central inflammation [223]. The critical role in the initiation and progression of neuroinflammation in Alzheimer’s disease is played by Aβ [244]. Neuroinflammation can be defined as “the presence of activated microglia and astrocytes which cause injury through the expression and release of proinflammatory cytokines, chemokines and complement increased generation of membrane fatty acids, eicosanoids, lipid peroxidation products and reactive oxygen and reactive nitrogen species” [245]. Astrocytes and microglia are involved in the innate immunity of the brain. Activation of microglial cells, which play an important role in central inflammation, may cause brain pathology [151,246]. Accumulation of Aβ stimulates increased inflammatory cytokines, chemokines and complementary proteins produced by chronically-activated glia [247]. Peripheral hyperinsulinemia, due to insulin resistance and accumulation of Aβ, causes increased cerebral levels of pro-inflammatory cytokines, such as interleukin-1β (IL-1β), interleukin-6 (IL-6) and tumor necrosis factor-α (TNF-α), which are associated with the inflammatory responses [248], all of which are detected in elevated levels in Alzheimer’s disease [243]. Also, in neuronal tissue of patients with diabetes mellitus and patients with Alzheimer’s disease, C-reactive protein is observed, an acute-phase reactant [249,250]. Research has shown that in peripheral insulin resistance, processes such as synthesis and release of pro-inflammatory cytokines and activation of inflammatory stress signaling may cause phosphorylation of insulin substrate-1 (IRS-1) by kinase inhibitor of kappa B kinase (IKK), c-Jun N-terminal kinase and extracellular signal-regulated kinase 2 (ERK2), resulting in interference with insulin receptor-mediated signaling. These processes cause a blockage of the intracellular action of insulin [251]. It is suggested that a similar mechanism occurs in the brain. According to this suggestion, Aβ plaques activate microglia, causing the secretion of pro-inflammatory cytokines. Secreted cytokines, binding to their respective receptors, activate the IRS-1 serine kinases [252]. Increased levels of pro-inflammatory cytokines, observed, as mentioned earlier, in patients with T2DM and patients with AD, can disturb insulin signaling in the brain [253].

### 4.4. Associations of Type 1 Diabetes Mellitus with Alzheimer’s Disease

There are very limited studies and inconsistent findings on the association between type 1 diabetes mellitus (T1DM) and the risk of developing Alzheimer’s disease. Therefore, the associations mentioned above remain uncertain [254]. Whether T1DM contributes to the risk of Alzheimer’s disease is still an unresolved question. Based on previous research, it is suggested that T1DM expedites cognitive decline [255]. Increased levels of phosphorylated tau protein were detected in patients with T1DM [256]. As mentioned above, this protein is associated with increased intracellular neurofibrillary tangle formation in patients with Alzheimer’s disease [255]. Animal studies showed that T1DM and AD have similar patterns of peripheral neuropathy in animal models. Researchers have investigated insulin signaling in the sciatic nerve in transgenic mice which over-express amyloid precursor protein and insulin-deficient mice. The results obtained revealed deficits in the insulin signaling pathway in both types of mice [257]. Another study was performed on neonatal mice. Researchers investigated associations of early peripheral sensimotor neuropathy with experimental diabetes in mice. As an index to measure the extent of peripheral neuropathy, the sciatic nerve and tibial nerves were used. Results revealed that the number of myelinated fibers, and their thickness and size, were decreased. Also reduced was the conduction velocity in nerves. Based on these results, researchers suggest that T1DM induces only peripheral neuropathy [258]. It has also been postulated that hyperglycemia and oxidative stress change the synaptosomal membrane fluidity and activity of membrane-bound enzymes. Results obtained suggest disturbances in membrane fluidity and composition of lipids, which may be a major factor in the development of diabetic encephalopathy [259]. Associations between T1DM and AD were also investigated using a bidirectional two-sample Mendelian randomized (MR) study [254]. The results obtained showed no significant correlation between genetically determined type 1 diabetes mellitus and Alzheimer’s disease (odds ratio (OR) = 0.984, 95% confidence interval (CI): 0.958–1.011, *p* = 0.247). According to those researchers, the results from other methods were different. MR Egger: OR = 0.957, 95% CI: 0.920–0.996, *p* = 0.039; weighted median: OR = 0.955, 95% CI: 0.928–986, *p* = 0.006. On the other hand, a final reverse MR study between Alzheimer’s disease and type 1 diabetes mellitus—AD: OR = 1.010, 95% CI: 0.991–1.116, *p* = 0.881. These results were consistent with other mentioned methods—MR Egger: OR = 1.023, 95% CI: 0.899–1.155, *p* = 0.716; weight median: OR = 1.014, 95% CI: 0.918–1.111, *p* = 0.758; weight mode: OR = 1.007, 95% CI: 0.907–1.108, *p* = 0.887. Based on the results obtained, which indicate no significant association between genetically determined T1DM and AD, the authors suggest that type 1 diabetes mellitus may play another, specific role in the development of neurodegenerative diseases in comparison with type 2 diabetes mellitus [254].

## 5. Diabetes Mellitus and Parkinson’s Disease

Parkinson’s disease (PD) is the second most common neurodegenerative disease. Its prevalence increases in elderly populations; approximately 1% of people aged >65 years are affected, and around 4% of those older than 85. It is treated as a late-life neurodegenerative disorder presenting different symptoms. The first medical description of Parkinson’s disease was performed in 1917 by James Parkinson. The four major symptoms characteristic of PD are rigidity, postural imbalance, bradykinesia and tremor. Also, a common symptom of PD is cognitive impairment and devastating non-motor disturbance. Research shows that in 25–46% of patients in the early stages of Parkinson’s disease, mild cognitive impairment may be detected that causes poor quality of life and may be involved in dementia in PD [260]. There are also other non-motor predictors for dementia, such as baseline severity of motor impairment, autonomic dysfunction, REM sleep disorder, hallucinations, depression, disturbances in vision or color, social phobia, anxiety and loss of smell [261,262,263]. Motor abnormalities, as mentioned earlier, include resting asymmetric tremors in the upper limbs, muscular rigidity, bradykinesia and gait dysfunction [186,264]. Motor abnormalities are due to progressive loss of the pigmented neuronal cells in the substantia nigra pars compacta (SNpc), which secrete dopamine. Dopamine, as a major neurotransmitter, is involved in the transmission of motor signals from the brain to the motor center. In PD, the level of dopamine is decreased [265]. A degeneration of projecting nerve fibers to the striatum is also observed in patients with PD [10]. The pathophysiological mechanism of PD is still not well understood, and multiple factors may be involved in the nigrostriatal pathway degeneration [266]. The diagnosis of Parkinson’s disease still depends on clinical manifestations [264]. Research shows that most diagnosed cases of PD are idiopathic [267]. There are also genetic and environmental factors involved in the development of PD [10]. For example, in Portugal, LRRK2 mutation was detected in 16.1% of patients with a familial form of PD and in 3.7% of patients with a sporadic form [268]. It was found that about 5–15% of PD cases are associated with genetic variants [269]. Research shows that familial forms of PD are associated with 23 loci and 19 genes [270]. As mentioned earlier, there are two forms of PD: the majority of cases of PD are sporadic (sPD), but several genes associated with familial forms of PD (fPD) have been identified [271].

### 5.1. Associations of Type 2 Diabetes Mellitus with Parkinson’s Disease

Research shows that patients with PD are more likely to develop several other diseases, such as cancer [272], cardiovascular disease [273], peripheral neuropathy [274], and others [275]. It has also been suggested that these patients may develop type 2 diabetes mellitus [276]. On the other hand, results of epidemiological studies are controversial [277]. According to several observations, there are suggestions that T2DM may be a risk factor for developing PD, but only in patients with long T2DM duration, for example over 10 years [278]. Note that many studies have revealed growing evidence that diabetic patients have an increased risk of developing PD [279]. It is very important to note that these diseases show similar dysregulated pathways. Therefore, there are suggestions of common underlying pathological mechanisms [280,281]. For example, T2DM is strongly associated with insulin resistance, which causes several disturbances in metabolism and inflammation. The pathogenesis of sporadic PD is associated with disturbances in glucose and energy metabolism [279].

The first step towards understanding the association between T2DM and PD is analysis of results obtained in epidemiological studies. There are many studies on this subject. For example, a meta-analysis performed in 2011 revealed increased risk in diabetic patients for developing Parkinson’s disease—relative risk RR = 1.37, 95%CI: 1.21–1.55, *p* < 0.0001 in prospective studies [282]. Results of other meta-analyses of population-based cohort studies showed a 38% increased risk in developing PD in diabetic patients—RR = 1.38, 95%CI: 1.18–1.62, *p* < 0.0001. Results obtained in this study revealed a higher risk of PD when T2DM duration was under 10 years and was higher in women—for females’ RR = 1.50, 95%CI: 1.07–2.11; for males’ RR = 1.40, 95%CI: 1.17–1.67 [283]. Other studies also observed an increased risk of developing PD in female patients, especially in subjects aged 40–79 years—odds ratio OR = 1.71, 95%CI: 1.60–1.82, *p* < 0.001; RR = 1.17, 95%CI: 1.11–1.30. These values for women older than 80 years were: OR = 1.39, 95%CI: 1.33–1.46, *p* < 0.001; RR = 1.09, 95%CI: 1.01–1.18 [284]. Based on these results, it is suggested that this age-dependent dynamic may be due to the protective role of sex steroids, such as 17β-estradiol on PI3-K/AKT and MAPK/ERK pathways [277]. On the other hand, in older patients (over 65 years old), no significant association was found between T2DM and PD—OR = 1.89, 95%CI: 0.90–3.98, *p* = 0.09. It was also found in this study that there is a significant positive association in diabetic patients with long-duration T2DM [278]. Other results revealed that patients with T2DM have a 30% higher risk of PD, as compared to patients without diabetes mellitus—RR = 1.30, 95%CI: 1.14–1.48 [285]. There are also many other analyses which confirm an association of T2DM with PD [177,266,277,279,286,287,288,289,290]. There are also other important observations: in patients with PD and T2DM, higher motor scores (*p* < 0.01), lower striatal dopamine transport binding (*p* < 0.05) and higher tau CSF levels (*p* < 0.05) were observed in comparison to patients with PD without T2DM [291]. T2DM in patients with PD causes faster progression of motor symptoms—Standardized mean different SMD = 0.55, 95%CI: 0.39–0.72 and cognitive decline SMD = 0.92, 95%CI: 1.50–0.34 [289]. Other observations revealed no difference in motor progression between patients with PD and T2DM in comparison with PD patients without T2DM, while T2DM causes faster cognitive decline in PD patients—OR/RR = 1.92, 95%CI: 1.45–2.55 [292]. In mildly diabetic patients with PD, atrophy in different parts of the brain was observed, such as cortical grey matter, amygdala, frontal white matter, and temporal white matter, as well as higher total white matter hypersensitivity and periventricular hyperintensities longitudinally. T2DM causes a greater rate of cognitive decline due to higher atrophy of white matter [260,264].

### 5.2. The Pathogenesis of Parkinson’s Disease

In patients with PD a global loss is observed in cerebral uptake as compared to healthy controls [293]. In patients with dementia, this loss in uptake was more severe. Uptake of glucose and blood flow were reduced on the contralateral side of the frontal cortex to the side of the body with muscle deficits in patients with unilateral disease [293]. Another study showed hypometabolism of glucose in PD patients with mild to severe cognitive impairments in the parietal, occipital, temporal and frontal lobes [294]. There was a correlation found between temporal hypometabolism of glucose in the parieto-occipital lobe with poor performance in cognitive assessment [295]. It is suggested that, for best understanding of these disturbances, observations on the expression of glucose transporters and activity of enzymes associated with glycolysis in PD should be included [186]. As mentioned earlier, neuronal loss in PD patients is often observed in specific areas of the substantia nigra and loss of the dopaminergic neurons with subsequent depigmentation. Intracellular accumulation of protein α-synuclein (α-Syn) and the presence of Lewy bodies [296,297] has also been detected in PD patients. Alzheimer’s disease and Parkinson’s disease are clinically distinct. Observations revealed a significant occurrence of Lewy bodies and Lewy neurites in the brains of patients with AD. Also, in amyloid plaques of brains in patients with AD, α-Syn was detected, and also in increased levels in cerebrospinal fluid. It is also suggested that tau and Aβ may have a role in Parkinson’s disease. Their presence in cerebrospinal fluid is used as an early diagnostic marker of Parkinson’s disease [298,299,300]. The α-syn protein monomer (α-Syn) plays a role as the precursor of aggregated α-syn forms. Its presence is detected in different brain cells, such as dopaminergic and noradrenergic neurons, microglial and astrocytes, as well as in different brain regions, such as the frontal cortex, hippocampus, striatum and olfactory bulb, where it is involved in many physiological functions [301]. Studies show that only the dopaminergic neurons in the pars compacta of the substantia nigra have increased vulnerability to oxidative stress, resulting in α-syn-associated pathology [302,303]. As mentioned earlier, there are many monogenic types of familial PD. Most cases of Parkinson’s disease are sporadic, and their pathogenesis is poorly understood [304]. The protein, as mentioned above, is located in the brain in most neurons, in their cytoplasm and organelles. It is involved in the regulation of processes such as mitochondrial fusion-fission, and prevents the import of cytosolic protein across the outer mitochondrial membrane to intermembrane mitochondrial space, promotes proteins with the SNARE motif to facilitate exocytosis, as well as interacting with clathrin to stimulate the formation of endocytic vesicles [302,305]. Intraneuronal locations of the α-syn are the synaptic terminals, nucleus, mitochondria, endoplasmic reticulum and the endolysosomal system [302]. It was found that overexpression of normal α-syn contributes to the formation of toxic α-synO and fibrillary conglomerates [302] and may cause pores in the cell’s plasma membrane [306,307], resulting in increased diffusion of Ca^2+^, which can result in cell death [308,309]. Exosomes in the calcium-dependent mechanism secrete toxic α-synO, resulting in brain transmission of Parkinson’s disease pathology from cell to cell [310]. Toxic α-syn forms contribute to several pathological processes, such as increased oxidative stress [311], impaired axonal transport [312] and disruption of the ubiquitin-proteasome complex [313]. These toxic forms also impair mitochondrial and synaptic function [314], inhibiting of the histone deacetylase acetylation of DNA and methylation by methyl transferase, impair transcription of DNA [315], impair trafficking and maturation proteins of the Golgi apparatus and transport Ca^2+^ associated with the Golgi apparatus [316,317]. Toxic α-syn forms also cause inhibition of vesicular traffic between the endoplasmic reticulum and Golgi apparatus due to the accumulation of undigestible α-syn forms in the endosomal system [318]. It interacts with the Kir6.2 subunit of the ATP-sensitive potassium channel and insulin-secretory granules in β-cells. This toxic form also causes downregulation of the secretion of insulin stimulated by glucose. Influencing of the ATP-sensitive potassium channel inhibits secretion of dopamine in the brain [290,319,320]. It was observed that α-syn can bind to other proteins, such as tau protein and β-amyloid, forming Lewy bodies (LBs) which are pathological neuronal inclusions of more than 70 proteins with core α-syn protein fibrillary aggregates [177,321,322,323]. The initial sites of LBs occurrence are cholinergic and monoaminergic brainstem neurons and neurons in the olfactory system. With progression of the disease, LBs are also found in limbic and neocortical brain regions [324,325]. Increased risk of the development of PD pathology may be due to brain insulin resistance, which causes attenuation of the insulin-degrading enzyme associated with inhibition of α-syn fibril formation from α-synO [326].

### 5.3. The Role of Type 2 Diabetes Mellitus in the Development of Parkinson’s Disease

As in the case of Alzheimer’s disease, pathologies associated with T2DM are involved in pathologies observed in PD. There are links between pathologies, such as insulin resistance, increased oxidative stress, hyperglycemia, neuroinflammation and disturbances in glucose metabolism (Figure 2).

#### 5.3.1. Insulin Resistance and Parkinson’s Disease

As mentioned before, insulin resistance contributes to pathologies, such as hyperglycemia, hyperinsulinemia, and hyperlipidemia, which are associated with pathogenesis of T2DM. Research shows a high prevalence of insulin resistance in patients with PD. In this group, impaired glucose tolerance is observed in 50–80% of patients [327]. Insulin is required for normal function of brain cells. The intracellular insulin signaling pathway is involved in the regulation of several processes. Research shows that patients with Parkinson’s disease have high insulin resistance and decreased levels of insulin receptor mRNA in SNpc in comparison with healthy controls [328,329]. Disturbances in the insulin signaling pathway in the frontal cortex may cause increased oxidative stress and accumulation of α-syn in PD [330]. Systemic insulin resistance due to T2DM is associated with the initiation, progression and accumulation of α-syn brain pathology associated with PD [331]. Insulin resistance in the brain is associated with α-syn brain pathology, due to stimulation of α-syn aggregation, as well as decreased clearance of α-syn by IDEs, autophagy and unfolded protein responses [332,333]. Human and animal studies have shown disturbances, such as the downregulation of the insulin receptor, endoplasmic reticulum stress, increased levels of intracellular ROS and chronic inflammation. The disturbances mentioned may have an impact on the insulin signaling pathway [65,333,334]. Insulin resistance accelerates PD progression and increases the likelihood of cognitive deterioration in patients with Parkinson’s disease [275]. It remains an unsolved problem: is insulin resistance a cause or an effect of PD [275]?

#### 5.3.2. Hyperglycemia and Parkinson’s Disease

Hyperglycemia observed in patients with T2DM also contributes to pathologies of PD. This action of hyperglycemia is due to several mechanisms, such as inhibition in SNpc synthesis, release and uptake of dopamine, microglial stimulation causing intense neuroinflammation, decreased expression of Parkin/PINK1 causing inhibition of mitochondrial function, and increased production of methylglyoxal (MGO), which is a very reactive dicarbonyl by-product released in processes of glucose metabolism. It causes glutamatergic hyperactivity and glycation of important cellular components, such as proteins, lipids and nucleic acids [301]. The effects of hyperglycemia are pathological, such as increased oxidative stress [335,336], the accelerated aggregation of α-syn [337], impaired mitochondrial function [338] and death of dopaminergic neurons [339]. These results were also confirmed by other authors in animal studies [340].

#### 5.3.3. Oxidative Stress and Mitochondrial Dysfunction and Parkinson’s Disease

Pathways of cellular energy production may be overactivated by hyperglycemia. For example, hyper-stimulated glycolysis and the Krebs cycle, which impair production of reduced Flavin adenine dinucleotide (FADH_2_) and nicotinamide adenine dinucleotide (NADH), can supercharge the electron transport chain that generates ROS. The main source of ROS in the brain are mitochondria in neuronal glia [341]. ROS play an important role in cell survival, but in this situation, their physiological level is important. ROS regulate several processes associated with survival pathways, such as antimicrobial, anti-inflammation and inhibition of tumors [106,342]. There are mechanisms which remove excess ROS associated with the antioxidant system. Unfortunately, in the central nervous system there is a lack of a powerful antioxidant system, and high consumption of oxygen is necessary and involves lipid components associated with lipid peroxidation. These characteristics suggest that the CNS is prone to damage due to ROS [177]. Research suggests a significant role of ROS in the loss of dopaminergic neurons [341]. Damage to neurons in the brain makes them permanently dysfunctional, as these cells are in a post-mitotic stage. Oxidative stress damages the substantia nigra, which contains a high population of dopaminergic neurons producing significant amounts of ROS [343]. Oxidative stress has several negative effects in cells, such as inactivation of enzymes, degradation of protein, and injury to DNA, resulting in irreversible damage [112,344]. It produces toxic products, such as quinone and semiquinone, caused by oxidation of dopamine and decreases activity of glucocerebrosidase, resulting in increased ROS levels [345,346]. Increased production of ROS activates several pathways, such as hexosamine pathways, formation of AGEs and PKCβ 1/2 [347]. There are high concentrations of polyunsaturated acids in the brain, which result in lipid peroxidation and the production of toxic products [348]. Excessive free radical generation, causing oxidative stress and production of cytokines, may be the main factors associated with cellular dysfunction, such as insulin signaling and mitochondrial dysfunction [275]. It is increasingly accepted that oxidative stress plays a pivotal role in the development of neurodegenerative diseases, as for example in the case of PD, causing the death of dopaminergic neurons [349].

#### 5.3.4. Neuroinflammation and Parkinson’s Disease

Insulin resistance, a symptom detected in patients with T2DM, causes mitochondrial dysfunction, resulting in the inflammation process. Previously, this process was described only as a response to injures, such as repair of tissues and defense against infections. Recent studies suggest that inflammation is an important factor associated with PD [350]. Increased intensity of chronic inflammation is correlated with the progression of motor and non-motor symptoms of Parkinson’s disease [301]. An initial event in the development of PD is neuroinflammation. Low-grade inflammation in a disease is common in the pathological mechanism. In response to the inflammatory process, microglial cells are recruited, which play a main role as a defensive system. They secrete pro-inflammatory cytokines, such as IL-2, IL-1β, IL-6, TNF-α, interferon-γ, CRP, as well as ROS and RNS [351,352]. In the case of brain insulin resistance, insulin receptors are downregulated on astrocytes and microglia. Impaired insulin signaling causes the increased secretion of inflammatory cytokines IL-6 and IL-8 by these cells [353]. Activation of microglia may have an initial protective effect. However, prolonged activation of these cells due to inflammation may increase apoptosis and neuronal death, and impair synaptic function [354]. It was also found that TNF-α and interferon-γ have neurotoxic effects on the central nervous system [177]. Results obtained in animal studies showed that increased microglial activation causes synaptic pruning of hippocampal nerve cells, causing impairment of memory. Microglial activation decreases microglial clearance of extracellular α-synuclein, and the resulting formation of α-synO forms and fibrils [197,355].

### 5.4. Associations of Type 1 Diabetes Mellitus with Parkinson’s Disease

As described in a previous section, various studies have been performed which confirm the role of T2DM in the development of PD. There have been far fewer investigations into the association between T1DM and Parkinson’s disease. Therefore, much less is known about this association, and our knowledge about this is very poor and unclear [356]. T2DM and T1DM are two entirely different disorders. To investigate the association between T1DM and PD, a two-sample Mendelian randomized (MR) statistical method was used: inverse variance weighting (IVW), MR-Egger, weight median and weighted mode. The results revealed that all methods used showed a lower risk of T1DM for developing PD. These results were as follows: IVW—OR = 0.93, 95%CI: 0.91–0.96, *p* = 3.12 × 10^−5^; MR-Egger—OR = 0.93, 95%CI: 0.88–0.98, *p* = 1.45 × 10^−2^; weight median—OR = 0.93, 95%CI: 0.89–0.98, *p* = 2.76 × 10^−3^; weighted mode—OR = 0.94, 05%CI: 0.90–0.98, *p* = 1.58 × 10^−2^. These findings were then replicated with another method (independent GWAS dataset on T1DM). The methods employed confirmed the results. Based on these results, the authors suggest that T1DM may have a protective effect against the risk of PD. But researchers also postulate that further studies are needed to clarify this mechanism [356]. Univariate conventional analysis performed in another study revealed the potential protective effect of T1DM on risk of developing PD (IVW—OR = 0.9708, 95%CI: = 0.9466–0.9956, *p* = 0.0214) [357]. Investigations of the risk of T1DM and anti-diabetic medications revealed that the correlation between T1DM and risk of PD was no longer significant [357]. A potential protective effect of T1DM on the risk and progression of PD was described also by other authors [358,359,360]. In an Austrian study, the results obtained on the association between T1DM and PD were as follows: 0R = 2.3, 95%CI: = 1.9–2.7 [361], and a GWAS study revealed a weak association between T1DM and PD [362]. Based on the results obtained, it is suggested that the higher risk in patients with T1DM for developing PD may be associated only with longer disease duration and not with other factors [266]. On the other hand, animal studies on a *Drosophila* model of T1DM revealed decreased cerebral levels of tyrosine hydroxylase, which is a PD-related phenotype [340]. This result may suggest an increased risk of PD due to T1DM. The results discussed come from several limited studies, which are sometimes different and controversial. Therefore, further investigations are needed to explore the association between T1DM and PD. Does T1DM cause a protective effect or an increased risk for developing PD? These questions and some others are very important, although it is necessary to explain the mechanism of the protective/negative role of T1DM on PD.

Note that PD increasing the blood sugar level may increase the risk of T2DM [363].

## 6. Diabetes Mellitus and Huntington’s Disease

Huntington’s disease is a progressive neurodegenerative disease. HD patients exhibit choreatic movements, psychiatric symptoms and cognitive decline. The most noticeable aspect of Huntington’s disease is an early onset. HD is usually diagnosed in people of 30–40 [364], and is inherited in an autosomal-dominant manner [364]. Huntington’s disease is an effect of mutation in the gene *HTT* that encodes the protein huntingtin. It is associated with unstable amplification of CAG (cytosine, adenine, guanine) and repeats in the gene in question [364,365]. The mutation is located in exon 1 of *HTT*. Expansion of CAG in the *HTT* gene causes synthesis of abnormally long huntingtin protein due to elongation of the polyglutamine tract. This form of huntingtin may have a tendency to aggregation [365]. The impaired protein is broken down by the cell into small fragments. These fragments are toxic; they aggregate and accumulate in neurons, resulting in HD. The aggregates of the mutant huntingtin (MHTT) may impair the process of transcription and control of protein quality. It is suggested that these pathological changes may be involved in impaired cognitive function and aberrant motor function, observed in patients with HD [365].

### 6.1. Associations of Type 2 Diabetes Mellitus with Huntington’s Disease

Research shows a higher prevalence of insulin perturbations in patients with HD. Oral glucose tolerance tests and intravenous arginine tolerance tests showed that 50% of patients have impaired insulin tolerance associated with hyperglycemia and hyperinsulinemia [366]. In other studies, 10.5% of patients with HD also had diabetes mellitus [367] and a third of patients with HD had impaired insulin tolerance too, while this value in the control group was 3% [368]. Changes in glucose metabolism and increased prevalence of T2DM in patients with HD were observed by other researchers [369,370,371].

### 6.2. The Pathogenesis of Huntington’s Disease and the Role of Type 2 Diabetes Mellitus in the Development of Huntington’s Disease

Disturbances in cellular glucose metabolism were also observed in the central nervous system of patients with HD. In symptomatic HD patients, the uptake of striatal glucose is decreased [372]. In other research, it was observed that reduced uptake of glucose in the caudate, putamen and thalamic regions is positively correlated with disease severity [373]. A loss in glucose uptake restricted to the frontal and inferior parietal cortex was detected in newly diagnosed patients. On the other hand, a global loss of glucose uptake was observed in patients with symptoms of HD lasting longer than 5 years. In another investigation, performed on symptomatic patients and carriers of asymptomatic mHTT, a decreased uptake of glucose in the caudate nucleus and putamen was noted, and this was correlated with the number of CAG repeats [374]. Investigations of post-mortem brain samples of patients with Huntington’s disease revealed a decreased expression of GLUT1 and GLUT3 by three- and four-fold, respectively. Similar observations in patients with earlier stages of HD showed that the expression of these glucose transporters was not changed [375]. Observations performed on animal models of HD showed a decreased uptake of glucose in the HD cortex, associated with the loss of glucose transport expression [186]. Based on animal models, it was suggested that disturbances in glucose metabolism in patients with HD may be due to impaired expression of the insulin gene. Use of animal models that express the *HTT* gene along with 140 CAG repeats revealed the development of impaired glucose tolerance at 8 weeks of age and diabetes mellitus by 18 weeks [376,377]. It was also found that the onset of glucoregulatory disturbance is similar in these animal models to the onset of symptoms of HD [377]. The precise mechanism involved in the decreased levels of insulin is not clear [378]. In patients with HD and in animal models, disturbances in mitochondrial metabolism were observed. The activity and expression level of complex II is reduced in HD patients and mouse models, so it is postulated that the activity of complex II may contribute to neuronal vulnerability in Huntington’s disease [379]. Post-mortem brain samples of HD patients also exhibited a loss of pyruvate dehydrogenase (PDH) and oxoglutarate dehydrogenase (OGDH), key enzymes in the tricarboxylic acid cycle [380,381]. As mentioned above, a decreased uptake of glucose and increased levels of lactate was noted in patients with HD. Based on this observation, it was suggested that HD causes decreased levels of energy [382]. More recent observations suggest that damage due to oxidative stress is associated with decreased expression of GLUT3, resulting in lactate build-up and inhibition of glucose uptake [383]. Research has shown that HTT plays an important role in mitochondrial dysfunction. The N-terminal of mHTT interacts with mitochondrial membranes, causing mitochondrial calcium disturbances, and mHTT can inhibit respiratory complex II. Changes in mitochondrial electron transport cause overproduction of ROS and decreased production of ATP [384]. According to the results presented above, excessive production of ROS and mitochondrial alterations were assumed in the pathogenesis of HD; however, which event occurs first remains elusive [385].

## 7. Diabetes Mellitus and Amyotrophic Lateral Sclerosis

Amyotrophic lateral sclerosis is a rare, progressive neurodegenerative disease. It is characterized by degeneration of the first and second motor neurons, causing spasticity and muscle atrophy. Motor neurons in the anterior horn of the spinal cord gradually diminish [386]. As a result, there may also be difficulties in speaking, swallowing and breathing, often causing death within a few years of diagnosis [364]. Two forms of ALS have been characterized: familial and sporadic, depending on the strength of the inherited genetic factor.

### 7.1. Associations of Type 2 Diabetes Mellitus with Amyloid Lateral Sclerosis

Research shows a protective role of diabetes mellitus in older patients, whereas in younger patients, DM increases the risk of developing ALS. Authors suggest that these differences are due to a different association of ALS with T1DM and T2DM [387,388]. A case-control study showed an association of T2DM with reduced risk of ALS and, 4 years later, its onset [389]. Most case-control studies from different countries revealed a decreased risk of developing ALS in patients with T2DM [390,391,392]. There are also studies revealing no significant effect on the development of ALS or progression of the disease, and sometimes revealing a higher risk of developing ALS in patients with T2DM aged below 65 years [393,394,395]. Another study showed a protective effect of T2DM against ALS only in patients aged 70 years or older [387,388]. The association of T2DM and ALS in dependence on age was described also in Asian population-based studies [396,397] and in a case-control study in the USA [389]. On the other hand, a retrospective cohort study revealed a greater risk of developing ALS in patients with DM [388], but this study was performed without distinguishing between T1DM and T2DM.

### 7.2. The Pathogenesis of Amyloid Lateral Sclerosis

Most cases are sporadic ALS, in which old age is an important risk factor. It usually appears in people aged 50–60 [364,398]. The cause of sporadic ALS is unknown, therefore it is difficult to research causal genes and environmental factors. In the case of familial ALS, in approximately 20% of patients the disease was due to mutation of the *SOD1* gene [399]. This gene is involved in several processes, such as post-translational modification, consumption of energy, and control of cellular respiration, as well as scavenging superoxide radicals [400]. Based on animal studies, it is suggested that, despite loss of antioxidant capacity due to mutation, mutated SOD1 is not associated with neurodegenerative disease [401]. On the other hand, other results suggest the gain-of-function of mutant SOD1 protein in motor neuron diseases [401]. In the yeast model of ALS, mutant SOD1 changes the biosynthesis of amino acids and induces cellular neural degeneration in ALS [402]. This enzyme converts O_2_^•−^ into hydrogen peroxide (H_2_O_2_) and molecular oxygen. Mutant SOD1 proteins cause increased generation of ROS, resulting in the death of motor neurons in ALS [403]. Oxidized or misfolded SOD1 causes mitochondrial dysfunction, leading to sporadic amyotrophic lateral sclerosis [404]. Alteration of signal transduction pathways in motor neurons and the activity of glial cells, due to mutation in SOD1, enhances the progression of familial ALS [403].

## 8. Comparison of Described Neurodegenerative Diseases and Associations with Diabetes Mellitus

The neurodegenerative diseases described—Alzheimer’s, Parkinson’s, Huntington’s disease and amyotrophic lateral sclerosis—are due to different factors, abnormal protein aggregation, hyperphosphorylation, mutations, etc. (Table 1). On the other hand, there are also similarities. These diseases are associated with diabetes mellitus, mainly with type 2 diabetes mellitus. In most cases, DM increases the risk of developing neurodegeneration, although there are cases in which T1DM protects against PD. T2DM may have a protective effect in the case of ALS, whereas T1DM increases risk for developing ALS. Diabetes mellitus is characterized by pathologies such as hyperglycemia, hyperinsulinemia, oxidative stress, and impaired function of mitochondria. These symptoms also revealed a negative effect in patients with particular neurodegenerative diseases (Table 2). Therefore, in the diseases mentioned, anti-diabetic drugs may be used as therapy.

## 9. Antidiabetic Drugs in Treatment of Neurodegenerative Diseases

The global prevalence of neurodegenerative diseases, such as AD, PD, HD, and ALS, is on the rise, primarily due to an aging population; this positions neurodegenerative diseases as a significant public health concern. Despite intensive research, few effective therapies which may prevent or delay the progression of neurodegenerative diseases have been developed. One of the well-defined risk factors for NDs is T2DM, and insulin resistance has also been proven to be related to cognitive decline. Certain antidiabetic drugs have shown promise in offering neuroprotective benefits and alleviating neurodegenerative disease symptoms. These drugs include, for example, metformin, glucagon-like peptide-1 receptor agonists, and peroxisome proliferator-activated receptor gamma agonists. The exact mechanisms of action of antidiabetic drugs in treating NDs remain elusive, but, based on results obtained to date, these drugs offer a promising novel strategy for managing cognitive disorders [405].

### 9.1. Metformin

Metformin is an antidiabetic drug from the family biguanides. It is widely endorsed as the first-line medication in T2DM therapy and exerts its primary hypoglycemic effect by improving insulin resistance and inhibiting hepatic gluconeogenesis [179]. Metformin plays a role as an insulin sensitizer, increasing cells’ sensitivity to insulin [406], but does not directly induce insulin secretion [266]. It has the ability to cross the blood–brain barrier (BBB) and penetrate to the brain tissue [266,407]. Metformin protects against oxidative stress and has an anti-inflammatory effect by reducing mitochondrial respiratory dysfunction by inhibiting mitochondrial respiratory chain complex 1 [408]. This insulin sensitizer decreases methylglyoxal (MG) levels in diabetic patients, working as a scavenger of MG [409]. MG is a glycolytic byproduct which results from the degradation of glycerylaldehyde-3-phosphate and dihydroxyacetone phosphate, in the process of catabolism of ketone bodies and glycated proteins. This byproduct may react with different proteins, forming irreversible modifications of AGE [266]. Metformin also activates the AMP-activated protein kinase (AMPK) pathway [408], which modulates cell stress and has prosurvival functions [277]. Research shows that metformin shows multiple beneficial effects in neurodegenerative diseases. It reduces toxic protein aggregates, promotes neurogenesis and enhances neuronal bioenergetics [364,410]. Experiments performed on animal models revealed significantly improved motor function, learning behavior and memory in high-fat diet-fed mice and rats after metformin treatment [411,412]. Clinical studies showed a positive influence of metformin on cognitive decline and Alzheimer’s disease [364]. Usage of metformin decreases the risk of cognitive impairment in T2DM [413]. The incidence of dementia is lower in diabetic patients receiving metformin in comparison to those not receiving oral anti-hyperglycemic agents [414]. In patients with T2DM receiving metformin, the risk of developing AD is lower as compared to diabetic patients receiving sulfonylureas and thiazolidinediones [415,416]. Metformin regulates Aβ and α-synuclein aggregation and phosphorylation of tau protein [364,417]. In mouse models of AD, it ameliorates Aβ deposition, which may be associated with strengthened expression of insulin-degrading enzyme or decreased expression and activity of *β*-secretase, an enzyme involved in the cleavage of the amyloid precursor protein required to generate Aβ [418,419,420]. By activating protein phosphatase 2A in the brain, metformin significantly increases the dephosphorylation of α-synuclein and tau [421,422]. It also reduces neuronal loss, rescuing cognitive deficits in AD. In this case, metformin protects against Aβ-induced apoptosis [423]. Neuroinflammation plays an important role in the development of AD and PD. Metformin significantly reduces the expression of pro-inflammatory cytokines, such as TNF-α and interleukin-1β [424]. On the other hand, there is evidence that metformin may cause cognitive decline and its prolonged use may cause an increased risk of AD [425,426]. The role of metformin in PD is controversial. In animal models of PD, metformin has beneficial effects. As mentioned earlier, it decreases MG levels and α-synuclein expression. In the substantia nigra and striatum, metformin ameliorates apoptosis of dopaminergic neurons and reduces the number of astrocytes and their hypertrophy [409]. Metformin controls dyskinesia development and neuroprotection [427]. But these observations revealed that metformin did not affect dopaminergic cell death. Human clinical studies have not looked solely at metformin. In these studies, effects of metformin were compared to, or in combination with, other anti-hyperglycemic drugs. There is a lack of clinical data suggesting a positive effect of metformin on PD risk [364]. Unfortunately, conflicting results have been published. For example, a recent meta-analysis revealed that metformin did not show any effect on NDs [428]. A significantly increased incidence of PD has also been reported when compared with patients who did not take metformin [277]. Further research is needed to elucidate the interplay of metformin in neurodegeneration. It was also found that male transgenic HD mice fed metformin over a long period of time had prolonged survival times in comparison with female transgenic HD mice [429]. Human studies showed that diabetic patients with HD receiving metformin had better cognitive test results as compared to HD patients without T2DM [430]. In the female mouse model of ALS, even with inhibition of estrogen production, metformin does not reduce the pathology of disease [431].

### 9.2. Thiazolidinediones

Thiazolidinediones (rosiglitazone, piaglitazone) are antidiabetic agents which increase insulin sensitivity of cells by binding to the proliferator-activated receptor gamma (PPARγ). PPARγ is expressed in the substantia nigra and putamen. It plays an important role in intracellular signaling pathways which regulate essential functions. In the basal ganglia, PPARγ can interfere, via peroxisome proliferator-activated receptor gamma coactivator 1-alpha (PGC1α), with mitochondrial biogenesis and with inflammation pathways, reducing oxidative toxicity in several animal models [432]. Observations performed on animal models revealed that pioglitazone binds to mitochondrial receptors modulating complex I and reducing oxidative toxicity. Thiazolidinediones exert neuroprotective effects in animal models of AD and PD, significantly improving behavior and motor responses [433,434,435]. Rosiglitazone improves motor deterioration and mutant huntingtin protein aggregation in mouse models of HD [436,437,438]. Neuroprotection of thiazolidinediones against cognitive impairment was confirmed in several experimental and clinical studies [439,440,441]. The results obtained suggest that pioglitazone has many properties which may be helpful in the treatment of Alzheimer’s disease [442]. Pioglitazone reduces the amount of Aβ deposits in in vitro studies on rat nerve cells, and inhibits the phosphorylation of PPAR-γ which regulates the expression of IDE. As mentioned earlier, IDE is an enzyme involved in the degradation of Aβ [443]. It decreases the concentration of TNF-α, a pro-inflammatory cytokine [444]. In patients with T2DM, Alzheimer’s disease and mild cognitive impairment (MCI), impairment in cognitive function was observed after pioglitazone therapy [445]. Results obtained in a meta-analysis revealed a significant decrease in PD in diabetic patients receiving thiazolidinediones [446], especially when follow-up duration was over 5 years. A dose-dependent benefit of pioglitazone was also reported [447]. In a transgenic ALS mouse model, pioglitazone delays the onset of ALS and significantly enhances the survival time of these mice [448]. On the other hand, in another study, pioglitazone failed to show significant effects in Parkinson’s disease in human trials [449]. Results obtained in animal studies revealed that rosiglitazone improves spatial memory and increases removal of Aβ deposits, by increasing IDE expression [450]. Unfortunately, there are many conflicting results from clinical trials regarding the efficacy of this drug for dementia therapy. Some observations suggest that this drug may improve cognitive performance in patients with MCI and AD [451]; however, other studies do not confirm this effect [452].

### 9.3. Glucagon-like Peptide-1 Receptor Agonists

Glucagon-like peptide-1 (GLP-1) is an endogenous multifunctional peptide released by enteroendocrine L cells after food intake. It induces pancreatic insulin secretion. GLP-1 is involved in regulation of blood glucose levels, food intake and body weight [453,454,455]. It is rapidly (within 2 min) degraded by dipeptidyl-peptidase 4 (DPP4). Diabetic patients release lower levels of GLP-1 in response to meals [266]. The GLP-1 receptor (GLP-1R) is mainly expressed in the intestine, α and β cells of pancreatic islets, and in many central nervous system regions. Its expression has been identified in the human hypothalamus, hippocampus, parietal cortex, insula and putamen [456]. GLP-1R regulates several functions, such as satiety, glucose homeostasis, stress response, synaptic plasticity and memory [457,458]. GLP-1R activates MAPK and PI3K/AKT signaling pathways, which are associated with anti-apoptotic actions and contribute to processes such as neurogenesis, inflammation and mitochondrial function [459,460]. GLP-1R agonists (GLP-1RAs), such as liraglutide, semaglutide, lixisenatide, albiglutide and dulaglutide have been endorsed for the clinical treatment of diabetic patients [405]. Studies both iv vivo and in vitro in different cell lines, animal models and patients with neurodegenerative diseases have revealed the neuroprotective effects of GLP-1RAs. Studies performed on animal models of AD showed that liraglutide increases neurogenesis and reduces the amount of Aβ and hyperphosphorylated tau, reducing inflammation. It has a positive effect on memory, due to participation in the formation of long-term potentiation (LTP) [461,462]. In a mouse model of AD, liraglutide improves memory function and increases the number of neurons in the hippocampus [463] and, in HD animal models, restores insulin sensitivity and increases cell viability. These actions in HD animal models are due to upregulation antioxidant pathways and reduction of oxidative stress [405]. In a streptozotocin rat model, liraglutide improved memory and anxiety, as well as reducing hippocampal neuronal death. On the other hand, it induced depressive-like behavior [464]. This agonist also induces cell survival in AD animal models [465]. In a rotenone model of PD, liraglutide and sitagliptin reversed nigral neuronal loss, reduced pro-inflammatory cytokines and improved motor performance [466]. As mentioned earlier, critical pathological events in AD and PD in animal models result from the aggregation of insoluble Aβ peptides, hyperphosphorylated tau and α-synuclein. Research suggests that GLP-1RA can ameliorate insoluble Aβ and α-synuclein aggregation and reverse tau phosphorylation [467,468]. GLP-1RAs are involved in mitochondrial function, enhancing the mitochondrial apoptotic pathway and reducing oxidative stress [469,470]. The role of GLP-1RAs in ALS remains less explored. Research suggests that liraglutide does not significantly alter lumbar spinal motor neurons or glial activation in a mouse model of ALS [471]. GLP-1RAs have shown positive effects in several animal models of NDs. However, it is suggested that the positive effects observed in animal models, need clinical research on the possible neuroprotective role of GLP-1R agonists in human subjects [266].

### 9.4. Dipeptidyl Peptidase-4 Inhibitors

Dipeptidyl peptidase-4 inhibitors (DPP4i) are widely used hypoglycemic agents, associated with low risk of hypoglycemia [472]. These drugs improve glucose metabolism by enhancing the bioavailability of active GLP-1, due to inhibition of its degradation. These inhibitors decelerate the inactivation of GLP-1, increasing its peripheral levels. In animal PD models, DPP4i showed neuroprotective properties. In these animal models, saxagliptin improved motor performance and mitochondrial complex I activity and reduced levels of inflammation molecules [473]. Vildagliptin revealed neuroprotective and antioxidative effects [474]. Positive results have also been observed in animal models with linagliptin [475] and sitagliptin [476]. The positive effects of DPP4i observed in animal studies were obtained with higher doses than the ones used in human treatments, therefore, exploring the results is difficult [266,277]. It should be noted that conflicting results have emerged [477], as some authors have observed paradoxic negative effects on cognition [296]. Regarding data in humans, the incidence of Parkinson’s disease significantly decreased in individuals receiving DPP4i [446,478], as compared to other antidiabetic treatments. A beneficial effect of DPP4 inhibitors was observed in PD patients with T2DM, who showed higher baseline dopamine transporter availability and better performance compared to nondiabetic patients or patients with T2DM who did not receive DPP4i [479].

In the case of other antidiabetic drugs, such as sulfonylureas, SGLT-2 inhibitors, and GIP receptor agonists, further research is needed for the confirmation of their beneficial effects in neurodegenerative diseases.

## 10. Limitations

There are several limitations in this study. These limitations include the selection of more than 50% of the studies published in the last 5 years. This paper provides a non-exhaustive overview of the current knowledge of the role of diabetes mellitus in the etiology of neurodegenerative diseases. Some of these results were obtained in animal studies. I have not appraised these studies. It is worth noting that several of the results presented are controversial and contradictory. For example, results obtained from case-control studies suggest a reduced risk of ALS in patients with T2DM, but there are also results which revealed no significant effect on development of ALS in these patients, as well as results showing a higher risk of developing ALS in patients with T2DM depending on age. There are also very few investigations on associations between T1DM and neurodegenerative diseases.

## 11. Conclusions

This review has described the association between diabetes mellitus, especially type 2 diabetes mellitus, and selected neurodegenerative diseases. However, there are different, sometimes controversial, results, and it is clear that in the majority of cases, DM increases the risk of developing neurodegenerative diseases, such as AD, PD, HD, and ALS. Only in the case of ALS does T2DM seem to have a protective effect, and that only in older patients. On the other hand, T1DM has a negative role in the development of ALS. However, the described diseases have specific causes, and the development and progression of these neurodegenerative diseases are associated with pathologies observed in T2DM, such as hyperglycemia, hyperinsulinemia, insulin resistance, mitochondrial dysfunction, and so on. Another common feature of these diseases is impaired glucose uptake, causing decreased production of energy in the brain. However, these results should be verified by further studies. The precise mechanism by which T2DM stimulates development and progression of AD, PD, and HD, or protects in ALS, is still not fully understood. Therefore, our knowledge in some cases may be insufficient. Further investigation of the associations mentioned between DM and neurodegenerative diseases is needed for curative therapy and/or effective prevention for these diseases.

## Figures and Tables

**Figure 1 ijms-26-00542-f001:**
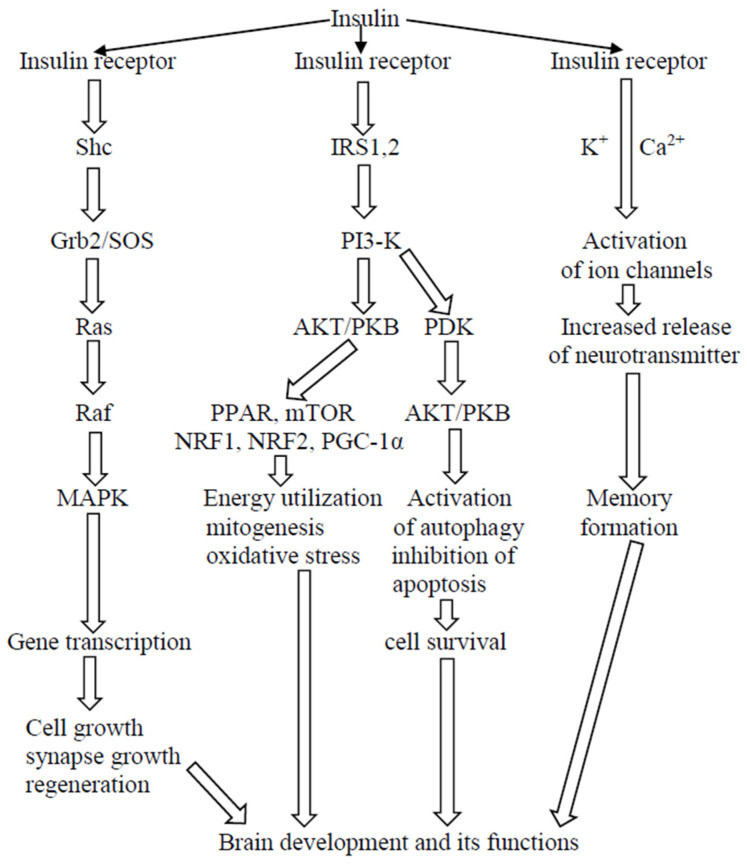
Insulin signaling pathway in a healthy human brain. Insulin signaling begins with the binding of insulin to an insulin receptor. Insulin receptor substrates differentiate the insulin signaling pathway into two main pathways: PI3-K–AKT and Ras–Raf–MAPK, which are involved in different processes [64]. AKT/PKB—protein kinase B, IRS1,2—insulin substrate 1 and insulin substrate 2, Grb2/SOS—growth factor receptor binding protein 2/Son of Sevenless protein, MAPK—mitogen activated protein kinase, mTOR—mammalian target of rapamycin, NRF1, NRF2—nuclear respiratory factor 1 and nuclear respiratory factor 2, PDK—phosphatidylinosite-dependent kinase, PGC-1α—peroxisome proliferator receptor γ coactivator 1-α, PI3-K—phosphatidylinositol 3 kinase, PPAR—peroxisome proliferator-activated receptor, Raf—regulation of alpha-fetoprotein, Ras—rat sarcoma virus peptide, Shc—Src homology collagen peptide.

**Figure 2 ijms-26-00542-f002:**
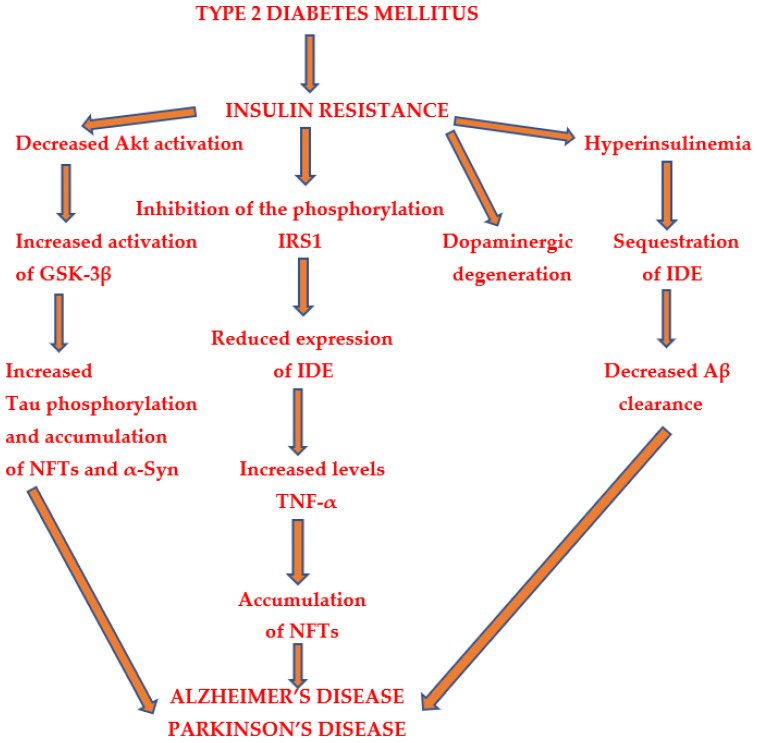
Associations of type 2 diabetes mellitus with Alzheimer’s and Parkinson’s diseases [177].

**Table 1 ijms-26-00542-t001:** The pathogenesis of the neurodegenerative diseases described above.

Neurodegenerative Disease	The Pathogenesis of Disease
Alzheimer’s disease	Alzheimer’s disease is caused by misfolded NFTs which contain hyperphosphorylated tau and aggregated Aβ peptides. Accumulation of Aβ impairs synaptic function. Tau protein in brains of patients with AD is three times more hyperphosphorylated as compared to healthy subjects. These pathologies are associated with pathologies observed in patients with T2DM, such as insulin resistance, hyperinsulinemia, hyperglycemia, oxidative stress, and disturbances in mitochondrial function. Research into the association between T1DM and AD showed no significant correlation between the diseases.
Parkinson’s disease	Parkinson’s disease is caused by intracellular accumulation of protein α-synuclein (α-Syn) and the presence of Lewy bodies in the brain. Hyperphosphorylated tau and Aβ may also be detected. Over-expression of normal α-syn contributes to the formation of toxic α-synO and fibrillary conglomerates, which impair the cell’s plasma membrane, and increased diffusion of Ca^2+^ into neurons, which causes their death. Therefore, neuronal loss of dopaminergic neurons is commonly observed in patients with PD. Toxic α-synO is involved in several pathological processes in neurons. It can bind to tau protein and Aβ, forming Lewy bodies.
Huntington’s disease	HD is caused by mutation of the gene *HTT* that encodes the protein huntingtin. The mutation is associated with unstable amplification of CAG repeats in the gene, causing synthesis of abnormally long huntingtin proteins that may have a tendency to aggregate. Impaired protein is broken down by the cell into toxic small fragments which aggregate and accumulate in neurons, resulting in HD.
Amyotrophic lateral sclerosis	The cause of sporadic ALS is unknown; however, it is associated with old age. Familial ALS is due to mutations in the *SOD1* gene, which is involved in several processes. Mutant SOD1 proteins induce cellular neural degeneration and death of motor neurons in ALS. Oxidized or misfolded SOD1 causes mitochondrial dysfunction, leading to sporadic amyotrophic lateral sclerosis, and impaired SOD1 enhances progression of familial ALS.

**Table 2 ijms-26-00542-t002:** The associations of diabetes mellitus with the neurodegenerative diseases described above.

Neurodegenerative Disease	Disturbances Due to Diabetes Mellitus
Alzheimer’s disease	Insulin resistance, observed in the case of T1DM and T2DM, impairs the insulin signaling pathway, causing enhanced phosphorylation of tau protein. It also decreases regional cerebral blood flow, causing decreased brain supply of oxygen, glucose and nutrients. Hyperglycemia may cause neurotoxicity, causing neuronal damage and neuropathy, and may also be involved in the increased risk of developing dementia, cognitive dysfunction and memory deficits and increased levels of hyperphosphorylated tau protein. Stimulation of microglia, due to hyperglycemia, causes these cells to release pro-inflammatory cytokines. Oxidative stress may impair cell survival and anti-apoptotic signaling. It also impairs energy balance in the brain. Oxidative stress also activates microglia and astrocytes which are involved in damage to protein and DNA.
Parkinson’s disease	Insulin resistance increases oxidative stress and accumulation of α-syn. It stimulates the progression of Parkinson’s disease and increases the likelihood of cognitive deterioration in patients with PD. Hyperglycemia increases oxidative stress, aggregation of α-syn and death of dopaminergic neurons. Oxidative stress causes inactivation of enzymes, damage to protein and DNA, resulting in irreversible damage. It plays an important role in the loss of dopaminergic neurons. Neuroinflammation is due to stimulation of microglial cells resulting in secretion of pro-inflammatory cytokines. Activated microglia are also involved in the formation of α-synO forms and fibrils.
Huntington’s disease	In patients with HD, glucose uptake is decreased due to decreased expression of GLUT1 and GLUT3 in the brain. Reduced uptake of glucose in the caudate, putamen and thalamic regions positively correlates with disease severity.
Amyotrophic lateral sclerosis	Different and sometimes controversial results have been obtained in studies to date. Therefore, the associations between DM and ALS require further investigation.

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
