# Peer review of "Associations Between Diabetes Mellitus and Neurodegenerative Diseases"

_ijms, 2025, doi:10.3390/ijms26020542_

Round 1
Reviewer 1 Report
Comments and Suggestions for Authors
The present manuscript entitled “Associations Between Diabetes Mellitus and Neurodegenerative Diseases” shows the connections between DM and neurodegenerative diseases. The present manuscript should be modified in the following ways.
1. Line 8: helath is not the appropriate word. Please revise the statement
2. Abstract can be improved by mentioning potential relations between DM and neurodegenerative diseases. Also can be mentioned (potential pathological events) where more research is required.
3. Add Neurodegenerative diseases in keywords.
4. Lines 138-156: Add references
5. Section 2: Most of the content in section 2 is not pertaining to Diabetes and its associated neurodegenerative diseases. It is about general information on diabetes pathophysiology and its types. Therefore, authors can reduce the content.
6. The pathophysiology of AD is well documented in the literature. Therefore, authors should focus on the links between DM and AD. Similar to PD.
7. Table 1 should be modified in such a way that it should address the potential links between DM and neurodegenerative diseases, but not the pathogenesis of neurodegenerative diseases.
8. There were several articles were published on the same topic. Further, the present manuscript is not different from the published articles. Therefore, there is a question about the novelty of the present work.
9. Several antidiabetic drugs have been shown to mitigate this neurodegenerative disease. For example, metformin, GLP1 agonists, gliptins, glitazones, and etc. However, the authors have not discussed the beneficial effects of these drugs against diabetes-associated neurodegenerative diseases.
10. Write the limitations of the present work.
11. Add 2-3 figures that show a potential correlation between DM and neurodegenerative diseases.
12. Overall the present manuscript lacks novelty.
13. Grammatical errors can be minimized.
Author Response
Dear Reviewer
Thank you very much for your opinion and suggestions. According to your suggestions, there were done changes and additional parts mentioned below:
- Line 8: helath is not the appropriate word. Please revise the statement
Response: I didn’t find word “helath” in line 8 and in next lines. I found word “health” - Abstract can be improved by mentioning potential relations between DM and neurodegenerative diseases. Also can be mentioned (potential pathological events) where more research is required.
Response: Abstract was changed according your suggestion, however, these changes are small because the number of words is limited. - Add Neurodegenerative diseases in keywords.
Response: In keywords is included “neurodegenerative diseases” and then are listed diseases described in manuscript. - Lines 138-156: Add references
Response: According to your suggestion, there were added references in lines 138-156 (Recent lines are 146-164). - Section 2: Most of the content in section 2 is not pertaining to Diabetes and its associated neurodegenerative diseases. It is about general information on diabetes pathophysiology and its types. Therefore, authors can reduce the content.
Response: The aim of Section 2 is the characterize of diabetes mellitus, therefore, there are described different pathologies associated with DM. I think that this, so rich section, is very important. Most people know only T1DM and T2DM, however, there are also other seven types of diabetes. Therefore, this section contains so many information. On the other hand, these information, especially on pathologies observed in DM are discussed in association with neurodegenerative diseases. It is also other cause, that is associated also with the next point. Maybe this article will be read not only by diabetologists and neurologists, but also by other specialists in medicine, as well as by specialists of other sciences, for example medical biologists. Therefore, I think that so many information is needed. - The pathophysiology of AD is well documented in the literature. Therefore, authors should focus on the links between DM and AD. Similar to PD.
Response: You’re right that the pathophysiology of Alzheimer’s diseases is well documented in literature. Also literature on this disease is very rich (also I wrote articles on AD). But the section of this issue is: Molecular Neurobiology, and Special Issue: Molecular Mechanisms and Treatments in Neurodegenerative Disease. Therefore, I think, that for better understanding links of DM and AD, PD, HD and ALS, is necessary to remind also general information on mentioned diseases. - Table 1 should be modified in such a way that it should address the potential links between DM and neurodegenerative diseases, but not the pathogenesis of neurodegenerative diseases.
Response: According to your suggestion, it is added table (Table 2) on association between DM and ND. - There were several articles were published on the same topic. Further, the present manuscript is not different from the published articles. Therefore, there is a question about the novelty of the present work.
Response: As I wrote above, there are several publications on DM in association with neurodegenerative diseases. But I think that is novelty my work. Firstly, several articles describe an association of DM with only one, sometimes two neurodegenerative diseases. In my article are described four neurodegenerative diseases. Very important is fact that these diseases are due to specific pathologies. My idea was: however, there are different causes of described diseases, but I all cases DM influences risk of diseases. Secondly, articles describe mainly association between T2DM with AD and PD, whereas is less analyses on HD and ALS. Thirdly, I wrote that there are more than 100 types of neurodegenerative disorders, of which about 20% is associated with DM. I wrote also that there are nine types of DM, not only T1DM and T2DM. This may be suggestion for next authors for description these associations with the remaining DM and neurodegenerative diseases. - Several antidiabetic drugs have been shown to mitigate this neurodegenerative disease. For example, metformin, GLP1 agonists, gliptins, glitazones, and etc. However, the authors have not discussed the beneficial effects of these drugs against diabetes-associated neurodegenerative diseases.
Response: You’re that several antidiabetic drugs are used in therapy of neurodegenerative diseases. My first idea was included also this section. But, as you see, an article is long. Mentioned subject is very rich in references and descriptions of performed investigations. Therefore, an addition of this chapter (and subchapters) may cause significant elongation of article and many additional references. I think that too long article and too rich in references, isn’t friendly for readers. Therefore, I resigned from this part, however, as I wrote, you’re right that this is very important problem. - Write the limitations of the present work.
Response: According to your suggestion, this information (Limitations) is added. - Add 2-3 figures that show a potential correlation between DM and neurodegenerative diseases.
Response: I think that additional figure(s) may cause manuscript more difficult for readers. Therefore, it is added one additional table (Table 2) on this subject. - Overall the present manuscript lacks novelty.
Response: About this suggestion, I wrote more in point 8. - Grammatical errors can be minimized.
Response: Proofreading was done by professional proofreader that cooperate with me from few years.
And again, Thank you very much for your suggestions. I’m sure that this article, if it will be accepted for publication, will be more easy and friendly for readers.
Reviewer 2 Report
Comments and Suggestions for Authors
The review titled " Associations Between Diabetes Mellitus and Neurodegenerative Diseases " is an interesting work, it can serve as a valuable reference in this field.
However:
Many sentences need references for example
References need to be added for this paragraph “It is to note, that elevated blood glucose levels also impairs glial cells, such as as- 73 trocytes, which are involved in maintaining brain glucose levels. This role of astrocytes is 74 due to neuron-astrocyte coupling. However, there are nine types of DM”
You have to review this paragraph it is not a good scientific sentence:
Diabetes mellitus is a metabolic disease, characterized by hyperglycemia. There are 86
nine types of DM. Prediabetes is a state when blood sugar is higher than it should be, but 87 not high enough to diagnose it as diabetes. Blood glucose levels in prediabetes are be- 88 tween 110 mg/dL and 125 mg/dL, (5.7–6.4 mmol/L). These patients are also defined by the 89 presence of impaired fasting glucose (IFG) and/or impaired glucose tolerance (IGT). In 90 patients with prediabetes is observed obesity, especially abdominal or visceral, 91 dyslipidemia with high triglycerides and/or low HDL cholesterol and hypertension [20].
References for “ In 101
these patients is diagnosed insulinopenia and are prone to diabetic ketoacidosis (DKA), 102however, without evidence of β-cell autoimmunity. This form of diabetes is strongly in- 103herited, but without association with HLA”
Regarding the content of the review
The review is very rich in information but it will be important to set the objective of the review to have a clear idea
The aim of this review is description of association between T1DM and T2DM and above-mentioned neurodegenerative diseases so you have to focus on T1DM and T2DM and minimize the text in 2. Types of Diabetes Mellitus
it is preferable to change the position of Wolfram Syndrome to the last type of diabetes
it will be 2.9 and Alström Syndrome will be 2.8.
You have to develop the Figure 1 to be more representative and more innovative
The part of 275 to 365 must be reviewed and shortened
The author should focus on the objective of the review because he develops too much pathological parts of degenerative diseases.
The author should use tables and figures to pass the idea because there is too much text in the review
The author must show the major revisions made in the text by highlighting the changes in a different colored text.
It is imperative to consider all these remarks to reinforce the manuscript's quality and conclude more accurately.
Author Response
Dear Reviewer
Thank you very much for your opinion and suggestions. According to your suggestions, there were done changes and additional parts mentioned below:
References need to be added for this paragraph “It is to note, that elevated blood glucose levels also impairs glial cells, such as as- 73 trocytes, which are involved in maintaining brain glucose levels. This role of astrocytes is 74 due to neuron-astrocyte coupling. However, there are nine types of DM”
Response: According to your suggestion, mentioned references are added.
You have to review this paragraph it is not a good scientific sentence:
Diabetes mellitus is a metabolic disease, characterized by hyperglycemia. There are 86
nine types of DM. Prediabetes is a state when blood sugar is higher than it should be, but 87 not high enough to diagnose it as diabetes. Blood glucose levels in prediabetes are be- 88 tween 110 mg/dL and 125 mg/dL, (5.7–6.4 mmol/L). These patients are also defined by the 89 presence of impaired fasting glucose (IFG) and/or impaired glucose tolerance (IGT). In 90 patients with prediabetes is observed obesity, especially abdominal or visceral, 91 dyslipidemia with high triglycerides and/or low HDL cholesterol and hypertension [20].
Response: In this paragraph were changed sentences, especially order of sentences and their association.
References for “ In 101
these patients is diagnosed insulinopenia and are prone to diabetic ketoacidosis (DKA), 102however, without evidence of β-cell autoimmunity. This form of diabetes is strongly in- 103herited, but without association with HLA”
Response: Suggested references are added.
Regarding the content of the review
The review is very rich in information but it will be important to set the objective of the review to have a clear idea
Response: The aim of this review is presented in Abstract and Introduction. My idea was description of associations of T1DM and T2DM with four neurodegenerative diseases. Two of these diseases are more frequent, whereas two reminders are less frequent.
The aim of this review is description of association between T1DM and T2DM and above-mentioned neurodegenerative diseases so you have to focus on T1DM and T2DM and minimize the text in 2. Types of Diabetes Mellitus
Response: The aim of Section 2 is the characterize of diabetes mellitus, therefore, there are described different pathologies associated with DM. Type 2 DM is diagnosed in more than 90% of patients with diabetes. Therefore, T2DM is more important as health problem. It is also important that there is more rich list of references on T2DM than on T1DM. I think that this, so rich section, is very important. Most people know only T1DM and T2DM, however, there are also other seven types of diabetes. Therefore, this section contains so many information. On the other hand, these information, especially on pathologies observed in DM are discussed in association with neurodegenerative diseases. It is also other cause, that is associated also with two of the next points. Maybe this article will be read not only by diabetologists and neurologists, but also by other specialists in medicine, as well as by specialists of other sciences, for example medical biologists. Therefore, I think that so many information is needed.
it is preferable to change the position of Wolfram Syndrome to the last type of diabetes
it will be 2.9 and Alström Syndrome will be 2.8.
Response: This change of position was done according to your suggestion.
You have to develop the Figure 1 to be more representative and more innovative
Response: I would like to, especially, that this figure was easy and friendly for readers. Therefore, it is so “poor”. But, write me please more details on your suggestion. It will be help for me.
The part of 275 to 365 must be reviewed and shortened
Response: Mentioned part, recent 412-502, is a part of chapter entitled “Disturbances associated with diabetes mellitus”. It describes, in short, types of insulin resistance, hyperglycemia and oxidative stress. This information is then used in description of influence of pathologies observed in DM on ND, as well as mechanisms of this influence. Therefore, this part is rich. I think that presented details of pathologies in DM may cause that understanding of these associations and mechanisms will be more easy. Therefore, there are included so many information.
The author should focus on the objective of the review because he develops too much pathological parts of degenerative diseases.
Response: The aim of reviewed article is description of associations between diabetes mellitus and neurodegenerative diseases. Chapter of DM contains many information, I think that necessary, on its pathologies. Therefore, I think that is also necessary to write more, however important, information on discussed neurodegenerative diseases. Therefore, the pathologies of neurodegenerative diseases are so in details described. I think also, that “value” of pathologies of DM, should be similar to “value” of pathologies in neurodegenerative diseases.
The author should use tables and figures to pass the idea because there is too much text in the review
Response: Too many figures and tables cause that an article is difficult for readers. Therefore, according to your suggestion, it is added, however only one, table (Table 2). This table contains information on associations of DM and ND. I think that in this case, table is better than figure. It is more easy.
The author must show the major revisions made in the text by highlighting the changes in a different colored text.
Response: I think that reviewed manuscript is after major revision. All changes are highlighting. It was also done proofreading that was done by professional proofreader that cooperate with me from few years.
It is imperative to consider all these remarks to reinforce the manuscript's quality and conclude more accurately.
Response: I think and I hope that your suggestions were considered in manuscript. If, manuscript needs additional changes, write me please.
And again, Thank you very much for your suggestions. I’m sure that this article, if it will be accepted for publication, will be more easy and friendly for readers.
Round 2
Reviewer 1 Report
Comments and Suggestions for Authors
Authors have not addressed several issues given in previous comments. Please refer to comments 6, 8, 9, 11, 12, 13. Moreover, the response of authors for the previous comments are not justified.
Comments on the Quality of English LanguageEnglish to be corrected.
Author Response
Dear Reviewer
Thank you very much for your opinion and suggestions. According to your suggestions, there were done changes and additional parts mentioned below:
6. The pathophysiology of AD is well documented in the literature. Therefore, authors should focus on the links between DM and AD. Similar to PD.
The pathophysiology of AD is well documented – this is true. But, as I wrote previous the section of this issue is Molecular Neurobiology, and Special Issue: Molecular Mechanisms and Treatments in Neurodegenerative Disease. Therefore, I think, that for better understanding links of DM and AD, PD, HD and ALS, is necessary to remind also general information on mentioned diseases.
8. There were several articles were published on the same topic. Further, the present manuscript is not different from the published articles. Therefore, there is a question about the novelty of the present work.
According to Section of this issue and Special Issue, I think, that an article is rich in information on this subject. I can write, that one article contains many important information. There are articles on mentioned subjects, but they contain only one or two aspects on these associations.
9. Several antidiabetic drugs have been shown to mitigate this neurodegenerative disease. For example, metformin, GLP1 agonists, gliptins, glitazones, and etc. However, the authors have not discussed the beneficial effects of these drugs against diabetes-associated neurodegenerative diseases.
I have added paragraph (9) on this therapy.
11. Add 2-3 figures that show a potential correlation between DM and neurodegenerative diseases.
It is added one figure on association between T2DM and insulin resistance and AD and PD.
12. Overall the present manuscript lacks novelty.
I think that my suggestion on novelty is described in point 8.
13. Grammatical errors can be minimized.
As I wrote in previous suggestions, proofreading was done by professional proofreader (he is Englishman and English is his native language). Our cooperation is long-lasting. Therefore, I have problem, what to do? To date, I hadn’t never this problem.
And again, Thank you very much for your suggestions. I’m sure that this article, if it will be accepted for publication, will be more easy and friendly for readers.
Reviewer 2 Report
Comments and Suggestions for Authors
The review titled " Associations Between Diabetes Mellitus and Neurodegenerative Diseases " is an interesting work, it can serve as a valuable reference in this field.
However:
Unfortunately, the author did not make an effort to consider the major comments.
The review is very rich in information, but it is not well structured, so the reader loses the thread
When the author added a table, he put a lot of text in the table, it became like paragraphs!
Usually, the tables are more representative and easier to have a clear idea, so the author has added tables and figures in this review
The author has to reduce the text in all the review
He has to review the figure to be more representable with for example illustrations of the genes, proteins and tissues
The author has to make more effort to focus on the objective of the review
The author must show the major revisions made in the text by highlighting the changes in a different colored text.
It is imperative to consider all these remarks to reinforce the manuscript's quality and conclude more accurately.
Author Response
Dear Reviewer
Thank you very much for your opinion and suggestions. According to your suggestions, there were done changes and additional parts mentioned below:
Unfortunately, the author did not make an effort to consider the major comments.
In this version, there are added new information (figure, paragraph etc).
The review is very rich in information, but it is not well structured, so the reader loses the thread
When I wrote this (invited) article, my idea was: (1) detailed description of diabetes mellitus (types, pathologies, etc), (2) characteristics of selected neurodegenerative diseases (pathology of AD, role of particular pathologies observed in DM in development of AD, and then the same information on PD, HD, ALS). I thought that this is good form for description of mentioned associations.
When the author added a table, he put a lot of text in the table, it became like paragraphs!
Usually, the tables are more representative and easier to have a clear idea, so the author has added tables and figures in this review
Because this article is long and contains many information, I think that, for make an article easy, table should contain summary of particular paragraphs. Therefore, tables are like paragraphs.
The author has to reduce the text in all the review
I have problem with this suggestion. Because this article contains different information, it is so long. I looked, how to reduce the text, and I don’t see, how to do this reduction. For me, all sentences in article are important. I don’t know, what can I excluded.
He has to review the figure to be more representable with for example illustrations of the genes, proteins and tissues
Recent version contains 2 figures.
The author has to make more effort to focus on the objective of the review
I think that I wrote this in abstract, when I wrote, what is the aim of this article.
The author must show the major revisions made in the text by highlighting the changes in a different colored text.
It is imperative to consider all these remarks to reinforce the manuscript's quality and conclude more accurately.
All changes in text are highlighting (yellow and red).
Round 3
Reviewer 1 Report
Comments and Suggestions for Authors
Authors have addressed all the issues
Author Response
Dear Reviewer
Thank you very much for your recent decision.
Reviewer 2 Report
Comments and Suggestions for Authors
The review titled " Associations Between Diabetes Mellitus and Neurodegenerative Diseases " is an interesting work, it can serve as a valuable reference in this field.
However:
Primary, the author should know that we evaluate articles the same way even if they are part of an invitation so It is not necessary to mention that this is an invitation in the answers
Unfortunately, the author did not make any considerable effort to respond to the major comments.
The figures aren’t good for the level of this journal
The table is like a paragraph
The author has not reduced the text in all the review
The author must show the major revisions made in the text by highlighting the changes in a different colored text.
It is imperative to consider all these remarks to reinforce the manuscript's quality and conclude more accurately.
Author Response
Dear Reviewer,
Thank you very much again for your opinion.
I’d like to write my answer on your suggestions.
Primary, the author should know that we evaluate articles the same way even if they are part of an invitation so It is not necessary to mention that this is an invitation in the answers
You are right. I agree with your opinion. I also was many times a reviewer, including journals of MDPI. But, sometimes I had problem.
Unfortunately, the author did not make any considerable effort to respond to the major comments.
Suggestions and opinions of Reviewers are very important for me. Therefore, I wanted to include answers and, if it was possible, change manuscript according to Reviewer’s suggestions. But sometimes it was difficult, because there was controversy. Therefore, in few cases, I gave explanation instead change in manuscript.
The figures aren’t good for the level of this journal
The table is like a paragraph
According to your opinion about figures. I watched many articles with different figures. These figures were similar to my in reviewed manuscript. Moreover, I have publications in different journals of MDPI, such as: Szablewski L. “Insulin resistance: the increased risk of cancers: Curr. Oncol. 2023, 30; Szablewski L. “Changes in cells associated with insulin resistance” Int. J. Mol. Sci. 2024, 25, 2397; Pliszka M., Szablewski L. “Associations between diabetes mellitus and selected cancers” Int. J. Mol. Sci. 2024, 25, 7476; SÄ™dzikowska A., Szablewski L. “Insulin and insulin resistance in Alzheimer’s disease” Int. J. Mol. Sci. 2021, 22, 9987. In mentioned publications are included figures which are similar to my in recent publication. These figures were accepted. My idea is, that figure should be easy for readers and must help in understanding of publication. Therefore, these figures are so “primitive and easy”. Is my answer possible for your acceptation?
In the case of tables, I agree with you. Therefore, they are changed. I hope, that now they are much less “paragraph-like”.
The author has not reduced the text in all the review
With this point, however, very important, I had problem. As you see, you wrote in previous reviews, that text should be reduced. But, unfortunately, it is enlarged. There are two causes. 1) according to other reviews, it was suggested include additional paragraphs and information. 2) I made a test, with one paragraph, what will be when it will be reduced. Will be it more easy or more difficult. I think that in presented manuscript, there are mainly included information necessary for understanding of text. Therefore, as an effect of reduction of paragraph, was increased problem with understanding this paragraph. It was due to loss of deleted information. Therefore, excuse me, I had to come back to previous version.
The author must show the major revisions made in the text by highlighting the changes in a different colored text.
All changes are highlighting. Yellow “islets” in text – changes due to proofreader. All red points in text, and in figures are due by me. Both, Table 1 and Table 2 are due by me.
It is imperative to consider all these remarks to reinforce the manuscript's quality and conclude more accurately.
